# Targeting branched *N*-glycans and fucosylation sensitizes ovarian tumors to immune checkpoint blockade

Hao Nie[1], Pratima Saini[2], Taito Miyamoto[3], Liping Liao[1], Rafal J. Zielinski[1], Heng Liu[3], Wei Zhou [3], Chen Wang[1], Brennah Murphy[3], Martina Towers[1], Tyler Yang[3], Yuan Qi [4], Toshitha Kannan [5], Andrew Kossenkov[6], Hiroaki Tateno [7], Daniel T. Claiborne [3], Nan Zhang[3], Mohamed Abdel-Mohsen [2,8] ✉ & Rugang Zhang [1,3,8] ✉

Aberrant glycosylation is a crucial strategy employed by cancer cells to evade cellular immunity. However, it's unclear whether homologous recombination (HR) status-dependent glycosylation can be therapeutically explored. Here, we show that the inhibition of branched *N*-glycans sensitizes HR-proficient, but not HR-deficient, epithelial ovarian cancers (EOCs) to immune checkpoint blockade (ICB). In contrast to fucosylation whose inhibition sensitizes EOCs to anti-PD-L1 immunotherapy regardless of HR-status, we observe an enrichment of branched *N*-glycans on HR-proficient compared to HR-deficient EOCs. Mechanistically, BRCA1/2 transcriptionally promotes the expression of MGAT5, the enzyme responsible for catalyzing branched *N*-glycans. The branched *N*-glycans on HR-proficient tumors augment their resistance to anti-PD-L1 by enhancing its binding with PD-1 on CD8⁺ T cells. In orthotopic, syngeneic EOC models in female mice, inhibiting branched *N*-glycans using 2-Deoxy-D-glucose sensitizes HR-proficient, but not HR-deficient EOCs, to anti-PD-L1. These findings indicate branched *N*-glycans as promising therapeutic targets whose inhibition sensitizes HR-proficient EOCs to ICB by overcoming immune evasion.

Epithelial ovarian cancer (EOC) continues to be the most lethal gynecologic cancer in the United States[1]. High-grade serous ovarian cancer (HGSOC) accounts for >70% of EOC cases and is responsible for the majority of EOC-associated mortalities[2]. Approximately 50% HGSOCs exhibit defects in the homologous recombination (HR) DNA repair pathway caused by genetic or epigenetic inactivation of HR pathway genes such as *BRCA1* and *BRCA2*[3]. Notably, poly (adenosine diphosphate [ADP]–ribose) polymerase (PARP) inhibitors have been approved as maintenance therapy in recurrent HGSOC cases with HR-deficient phenotype, offering sustained clinical benefits[4,5]. However, a significant clinical challenge remains in establishing effective treatments for HGSOC with HR-proficient phenotype.

Immunotherapeutic agents bolster immune responses and/or overcome immune checkpoints, thereby restoring endogenous

[1]Department of Experimental Therapeutics, University of Texas MD Anderson Cancer Center, Houston, TX 77054, USA. [2]Vaccine and Immunotherapy Center, The Wistar Institute, Philadelphia, PA 19104, USA. [3]Immunology, Microenvironment and Metastasis Program, The Wistar Institute, Philadelphia, PA 19104, USA. [4]Department of Bioinformatics & Computational Biology, University of Texas MD Anderson Cancer Center, Houston, TX 77054, USA. [5]Bioinformatics Facility, The Wistar Institute, Philadelphia, PA 19104, USA. [6]Gene Expression and Regulation Program, The Wistar Institute, Philadelphia, PA 19104, USA. [7]Cellular and Molecular Biotechnology Research Institute, National Institute of Advanced Industrial Science and Technology (AIST), Tsukuba, Ibaraki 305-8566, Japan. [8]These authors jointly supervised this work: Mohamed Abdel-Mohsen, Rugang Zhang. ✉e-mail: mmohsen@wistar.org; rzhang11@mdanderson.org

antitumor immunity[6,7]. Despite the success of immunotherapies in certain cancer types, clinical trials have shown modest efficacy for these strategies, such as immune-checkpoint blockade (ICB) therapies, in EOC[8,9]. Consequently, there is an urgent need to discover effective immunological targets to fully harness the potential of immunotherapy for cancers such as EOC.

Recent advancements in the emerging field of glyco-immunology have revealed that several crucial immunological responses are mediated by glycans and their interactions with glycan-binding proteins (known as lectins). The recognition that glycans significantly modulate immunological functions has led to the realization that tumor cells employ aberrant glycosylation patterns to evade the host immune response[10,11]. For example, both branched- and core-fucosylated glycomic antigens have a significant association with cancer and immunity[12]. Lewis antigens, which contain branched fucose, on colon tumor cells, can bind to the C-type lectin DC-SIGN on macrophages and immature dendritic cells, thereby influencing the functions of these immune cells[13]. Core fucosylation, regulated by FUT8, is frequently upregulated in various cancers. Disrupting this core fucosylation process can impede PD-1/PD-L1-mediated immune evasion, thereby bolstering anti-tumor immunity[14]. Consequently, targeting fucosylated glycans on the glycoproteins of tumor cells has been recognized as a viable and promising strategy to amplify anti-tumor immune responses[15]. However, whether aberrant glycosylation can be leveraged as a genetic context-dependent vulnerability to develop cancer therapeutic strategies with precision has never been explored. Here, we show that elevated branched N-glycans are a strategy that HR-proficient tumors use to evade immune surveillance, and that the inhibition of branched N-glycans sensitizes HR-proficient, but not HR-deficient, EOCs to ICB.

## Results

### EOCs undergo glycomic alterations in response to immune pressures

To identify cell surface glycomic alterations that confer a survival advantage for EOCs under immune pressure, we injected the mouse HR-proficient UPK10[16] and HR-deficient HGS2[17] cell lines into the ovarian bursal sacs of both immunocompromised and immunocompetent mice (Supplementary Table 1). After 4 weeks, we harvested the tumors, sorted tumor cells by flow cytometry (FACS), and profiled the glycome of their cell-membrane proteins (Fig. 1a and Supplementary Fig. 1). For glycomic analyses, we employed the lectin microarray technology, which enables sensitive assessment of multiple glycan structures using a panel of immobilized lectins with known glycan-binding specificity[18–25]. Our glycomic profiling revealed that, regardless of their HR status, EOC cells grown in immunocompetent mice exhibited higher levels of total fucose compared to cells grown in the absence of immune pressure in immunocompromised mice. This increase in total fucose was evident through higher binding to a lectin specific for total fucose (*Aspergillus oryzae* (AOL) lectin)[26] (Fig. 1b).

Considering the pressing clinical needs to enhance immune responses against HR-proficient EOCs, we conducted injections of HR-proficient (ID8[27] and UPK10) and HR-deficient (BPPNM[3] and HGS2) cancer cells into immunocompetent mice (Fig. 1c and Supplementary Table 1). Subsequent glycomic profiling of the FACS-sorted tumor cells revealed that HR-proficient cells displayed a distinct glycomic signature compared to those observed in HR-deficient cells. Notably, among this glycomic signature, the cell membrane proteins of HR-proficient cells exhibited higher binding to two lectins, *Phaseolus Vulgaris* Leucoagglutinin (PHA-L) and *Phaseolus Vulgaris* Erythroagglutinin (PHA-E), which bind to branched *N*-glycans (Fig. 1d). These findings indicate that immune pressures drive specific glycan alterations such as increases in fucosylation on EOC cells, with HR-proficient EOC cells exhibiting unique glycomic signatures as exemplified by branched *N*-glycans.

### Inhibiting fucosylated glycans sensitizes ovarian tumors to anti-PD-L1 immunotherapy

Consistent with our findings, fucosylation has been extensively linked to cancer immune responses[12,28]. To validate our experimental approach, we initially focused on fucosylation, whose levels were elevated on the surface of EOC cells when they grew in immunocompetent compared with immunocompromised mice regardless of HR-status (Fig. 1b). The AOL lectin binds to a wide range of fucosylated structures, including branched and core fucose (Fig. 2a). These fucosylated glycans are catalyzed by various fucosyltransferases, such as FUT4, FUT8, and FUT9. Accordingly, we measured the expression of several of these fucosyltransferases in tumor cells grown in either immunocompromised or immunocompetent mice. Indeed, we found that the expression of several fucosyltransferases was significantly upregulated in tumors grown in immunocompetent mice compared to those grown in immunocompromised mice (Fig. 2b–d). These data provide confirmation that enhanced fucosylation represents a glycomic alteration driven by immune pressures on EOCs.

Subsequently, we investigated the impact of inhibiting this excessive fucosylation using fucose analogs, specifically 2-fluoro-L-Fucose (2FF)[29], on immune responses against EOCs both in the presence and absence of anti-PD-L1 immunotherapy (Fig. 2e). Notably, these treatments were well-tolerated and did not affect mice weight (Supplementary Fig. 2). While 2FF treatment alone did not significantly impact tumor growth, it significantly sensitized both HR-proficient (UPK10 and KPCA) and HR-deficient (BPPNM and HGS2) tumors to anti-PD-L1 immunotherapy (Fig. 2f, g). To have a better understanding of how 2FF enhance anti-PD-L1 effect on ovarian cancer, we conducted single-cell RNA sequencing to characterize the tumor immune microenvironment in mice treated with anti-PD-L1 alone and in combination with 2FF treatment (Supplementary Fig. 3). Signaling pathway enrichment analysis showed that compared to single anti-PD-L1 treatment, combination treatment induced dysfunction in genes related to CD8+ T cells, neutrophils, CD4+ T cells and regulatory T (Treg) cells (Fig. 2h, i). These findings confirm the validity of our approach, as they align with previous studies that have established a connection between inhibition of fucosylation and anti-tumor immunity[15,30,31].

### HR-proficient ovarian tumors exhibit elevated levels of MGAT5, the enzyme that catalyzes branched *N*-glycans compared to HR-deficient tumors

Given the critical unmet need to enhance therapeutic options specifically against HR-proficient tumors, we next focused on our findings that HR-proficient tumors exhibit higher levels of branched *N*-glycans compared to HR-deficient tumors (Fig. 1d). MGAT5 is the enzyme that is responsible for catalyzing the type of branched *N*-glycans we observed to be enriched on HR-proficient EOCs (Fig. 3a). Indeed, knockdown of MGAT5, but not MGAT3, in HR-proficient cell lines effectively blocked the synthesis of branched *N*-glycans, as indicated by reduced PHA-L binding (Fig. 3b and Supplementary Fig. 4). However, MGAT5 knockdown did not impact cell proliferation (Supplementary Fig. 5). These findings support the notion that MGAT5 expression drives the observed increase in branched *N*-glycan levels in HR-proficient compared with HR-deficient EOCs.

Notably, a Kaplan–Meier meta-analysis involving 373 patients with HGSOCs revealed that high expression of *MGAT5* mRNA was significantly associated with poor overall survival (Fig. 3c). To understand the mechanism underlying this observation, we compared the expression of MGAT5 in HR-proficient vs. -deficient mouse and human cell lines. Our results show that *MGAT5* mRNA is expressed at a higher level in HR-proficient tumors (UPK10 and ID8) compared to HR-deficient tumors (BPPNM and HGS2) (Fig. 3d). Consistently, MGAT5 protein levels are upregulated in a series of HR-proficient compared to HR-deficient EOC cell lines (both human and mouse-derived cell lines) (Fig. 3e). Notably, MGAT5 is downregulated by *Brca2* knockout in ID8

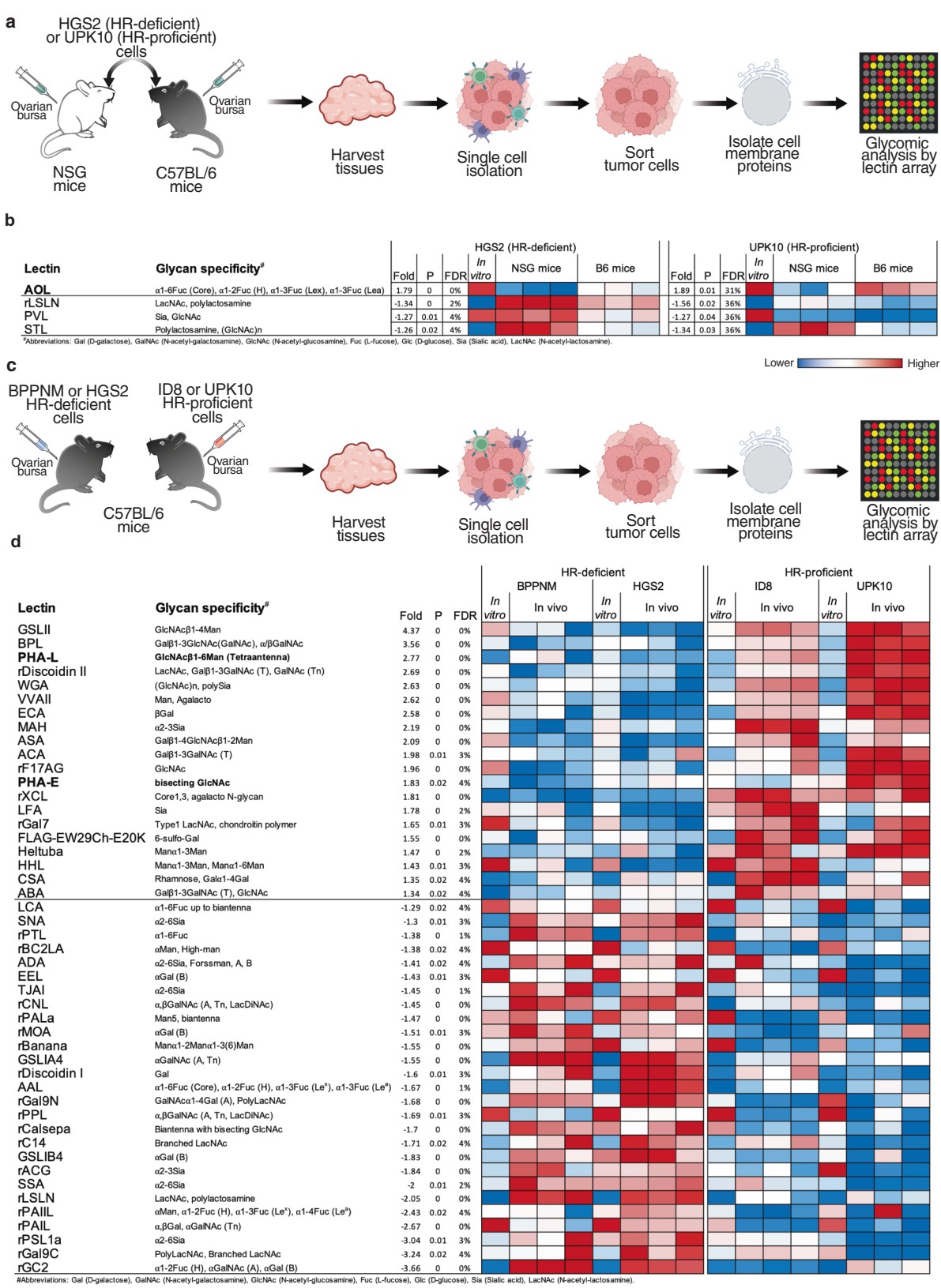

cells (Fig. 3e). Likewise, MGAT5 is expressed at a lower level in *Brac1* knockout BPPNM compared with the isogenic *Brca1* wildtype PPNM cell line (Fig. 3e). The observed decrease in MGAT5 induced by knockout of *Brca1* or *Brca2* is not a consequence of a decrease in S phase of cell cycle. For example, percentage of cells in the S phase of the cell cycle was not decreased by *Brca1* knockout in BPPNM cells

compared with *Brca1* wildtype PPNM cells (Fig. 3f). Likewise, *Brca2* knockout in ID8 cells (together with *Tp53* knockout) did not decrease the percentages of cells in the S phase of the cell cycle (Supplementary Fig. 6).

To validate these findings, we mined various databases for a correlation between the expression of *BRCA1/2* and *MGAT5*. In the

**Fig. 1 | Profiling of glycomic alterations in ovarian tumors under immune pressures. a** Schematic representation of experimental design examining the impact of immune pressures on ovarian cancer cell surface glycosylation. Components of the graphic were created with BioRender.com. **b** Heatmap depicting the levels of specific glycan structures in tumors grown in immunocompetent (C57BL/6; B6) or immunocompromised (NSG) environments. In vitro cancer cells were used to represent basal glycosylation. Red represents higher expression, while blue represents lower expression. $n = 3$ mice. *P*-values and False Discovery Rates (FDR) for the comparisons were estimated using R package limma v.3.46.0. **c** Schematic

representation of experimental design examining the differences between HR-proficient and HR-deficient ovarian tumors grown in B6 mice. Components of the graphic were created with BioRender.com. **d** Heatmap depicting the levels of specific glycan structures in HR-proficient vs HR-deficient tumors grown in immunocompetent B6 mice. In vitro cancer cells were used to represent basal glycosylation. Red represents higher expression, while blue represents lower expression. $n = 3$ mice. *P*-values and False Discovery Rates (FDR) for the comparisons were estimated using R package limma v.3.46.0.

Cancer Cell Line Encyclopedia (CCLE) database, we indeed observed a positive correlation between *MGAT5* expression and *BRCA1/2* expression in 1139 *BRCA1/2* wildtype cancer cell lines (Fig. 3g). A similar observation was made using the TCGA HGSOC dataset (Fig. 3h). Consistently, *MGAT5* amplification/overexpression is statistically mutually exclusive with genetic alterations that cause HR defects in the TCGA HGSOC dataset ($P < 0.036$; Fig. 3i). Finally, *MGAT5* is expressed at a significantly lower level in HGSOCs with driver mutations in HR pathway genes compared with those without mutations (Fig. 3j).

We next sought to determine the mechanism by which BRCA1/2 regulate MGAT5 expression. In addition to its canonical function in DNA repair, BRCA1/2 function as transcription activators in gene regulation[32–35]. Indeed, BRCA1 chromatin immunoprecipitation followed by next-generation sequencing (ChIP-seq) analysis from ENCODE[36] revealed that *MGAT5* is a direct target gene of BRCA1 (Fig. 4a). Consistently, ChIP-qPCR analysis confirmed direct binding of BRCA1 and BRCA2 to the *MGAT5* gene promoter in *BRCA1/2* wildtype EOC cell lines (Fig. 4b). This finding supports that notion that BRCA1 and BRCA2 directly promote the transcription of *MGAT5*. To functionally validate these findings, we knocked down BRCA1 or BRCA2 in HR-proficient cells. Indeed, knockdown of BRCA1/2 decreased MGAT5 expression both at the mRNA (Fig. 4c) and protein (Fig. 4d) levels. Consistently, we observed a reduction in PHA-L binding upon BRCA1/2 knockdown, indicating decreased synthesis of branched *N*-glycans (Fig. 4e). The reporter activity of the *MGAT5* gene promoter was significantly decreased upon BRCA1 knockdown (Fig. 4f-g). This activity was restored by the re-expression of the full length of BRCA1 (Fig. 4f-g); however, re-expression of a truncated BRCA1, lacking the transcription regulation domain BRCT1&2[37], failed to restore the reporter activity of the MGAT5 gene promoter (Fig. 4f-g). We next investigated whether BRCA1's DNA repair function is involved in promoting MGAT5 expression. To activate BRCA1's DNA repair function, DNA damage was induced by cisplatin in HR-proficient OVCAR3 cells. Indeed, we observed an increase in the formation BRCA1 foci (Fig. 4h), a marker of its activation in DNA repair[38]. Notably, MGAT5 expression is reduced in cisplatin-treated OVCAR3 cells (Fig. 4i). Thus, BRCA1's role in promoting MGAT5 expression is not due to activation of its role in DNA damage repair. These findings provide insights into the regulatory relationship between BRCA1/2 and MGAT5 expression, unraveling the connection between the HR pathway and the glycomic alterations observed in HR-proficient EOC cells.

### Branched *N*-glycans increase the binding affinity of PD-L1 on HR-proficient cancer cells to PD-1 on CD8[+] T cells

Our subsequent focus was to identify the downstream consequences of increased branched *N*-glycans on immune responses to HR-proficient EOC cells. To achieve this, we utilized the glucose/mannose analog 2-deoxy-d-glucose (2DG), a well-established tool for inhibiting branched *N*-glycans formation[39]. Treating human and mouse HR-proficient EOC cells with 1 mM 2DG for 48 h significantly reduced branched *N*-glycan and bisecting GlcNAc expression, as indicated by the reduction in PHA-L and PHA-E binding (Fig. 5a, b and Supplementary Fig. 7a). Since it was suggested that branched *N*-glycans could play a role in modulating the binding of PD-L1 to PD-1[39], we proceeded to examine the impact of 2DG and its subsequent

branched *N*-glycan inhibition on PD-L1's capacity to bind PD-1 upon IFNγ treatment (to stimulate PD-L1 expression). We observed that 2DG significantly reduced the ability of PD-L1[+] HR-proficient EOC cells to bind recombinant PD-1 (Fig. 5c). The same concentration of 2DG had no significant effect on cell proliferation (Supplementary Fig. 7b, c) or glycolysis (Supplementary Fig. 7d) within the same time frame. Notably, upon IFNγ stimulation, HR-proficient cells have a similar level of PD-L1 expression compared to HR-deficient cells (Supplementary Fig. 8a, b), but PD-1 binding to HR-proficient cells trends towards being higher than HR-deficient cells (Supplementary Fig. 8c, d), suggesting qualitive differences in HR-proficient cells that enhances their binding to PD-1. To confirm the link between branched *N*-glycans and T cell immune responses, we utilized the TISIDB dataset[40], which derives from the TCGA dataset and facilitates the exploration of tumor-immune interactions. Indeed, MGAT5 expression correlated with a decrease in infiltration of activated CD8[+] T cells in HGSOCs (Fig. 5d).

Next, we sought to examine whether branched *N*-glycans confer resistance to HR-proficient cancer cells against T cell-mediated killing. For this purpose, we utilized T cells expressing chimeric antigen receptors (CAR T cells) targeting human CD19 (hCD19)[41]. As CD19 is expressed at low levels on human EOC cells, we engineered HR-proficient PEO4 and OVCAR3 (Supplementary Fig. 9a, b), as well as the HR-deficient PEO1 (Supplementary Fig. 9c) cell lines to overexpress human CD19. Subsequently, we conducted killing assays with anti-CD19 CAR T cells at a 1:6 effector:target (E:T) ratio for 48 h. Our findings demonstrated that inhibiting branched *N*-glycans by pretreating cells with 2DG or knockdown of MGAT5, in combination with anti-PD-L1 treatment, significantly sensitized HR-proficient EOC cells to T cell assault, as measured by an elimination index reflecting the killing efficacy of CAR T cells (Fig. 5e, f and Supplementary Fig. 9d, e). Notably, 2DG pretreatment did not increase the killing efficacy in MGAT5 knockdown cells to a degree comparable to those observed in control cells (e.g. Fig. 5e). This supports the notion that the observed effects by 2DG pretreatment were due to inhibition of MGAT5 activity. As a control, pretreatment with 2DG in the presence of anti-PD-L1 did not impact the sensitivity of HR-deficient EOC cells to T cell-mediated cytotoxicity (Fig. 5g and Supplementary Fig. 9f). These results support that the elevated expression of branched *N*-glycans on HR-proficient EOC cells enhances the binding of PD-L1 on these cells to PD-1 on CD8[+] T cells. Additionally, inhibiting branched *N*-glycans is sufficient to overcome these detrimental consequences, which may sensitize HR-proficient EOCs to anti-PD-L1 immunotherapy.

### Inhibiting branched *N*-glycans enhances the sensitivity of HR-proficient ovarian tumors to anti-PD-L1 immunotherapy in vivo

We proceeded to investigate the potential of inhibiting branched *N*-glycans, using shMGAT5 or 2DG, on the response of HR-proficient EOCs to anti-PD-L1 immunotherapy in vivo, employing orthotopic syngeneic mouse models. 2DG treatment was well-tolerated and did not affect mice weight (Supplementary Fig. 10). Although MGAT5 knockdown or 2DG treatment alone did not affect tumor size, they significantly sensitized HR-proficient tumors (KPCA and UPK10) to anti-PD-L1 treatment (Fig. 6a–c). This finding further reinforced the connection between branched *N*-glycans and HR-proficient EOCs, as

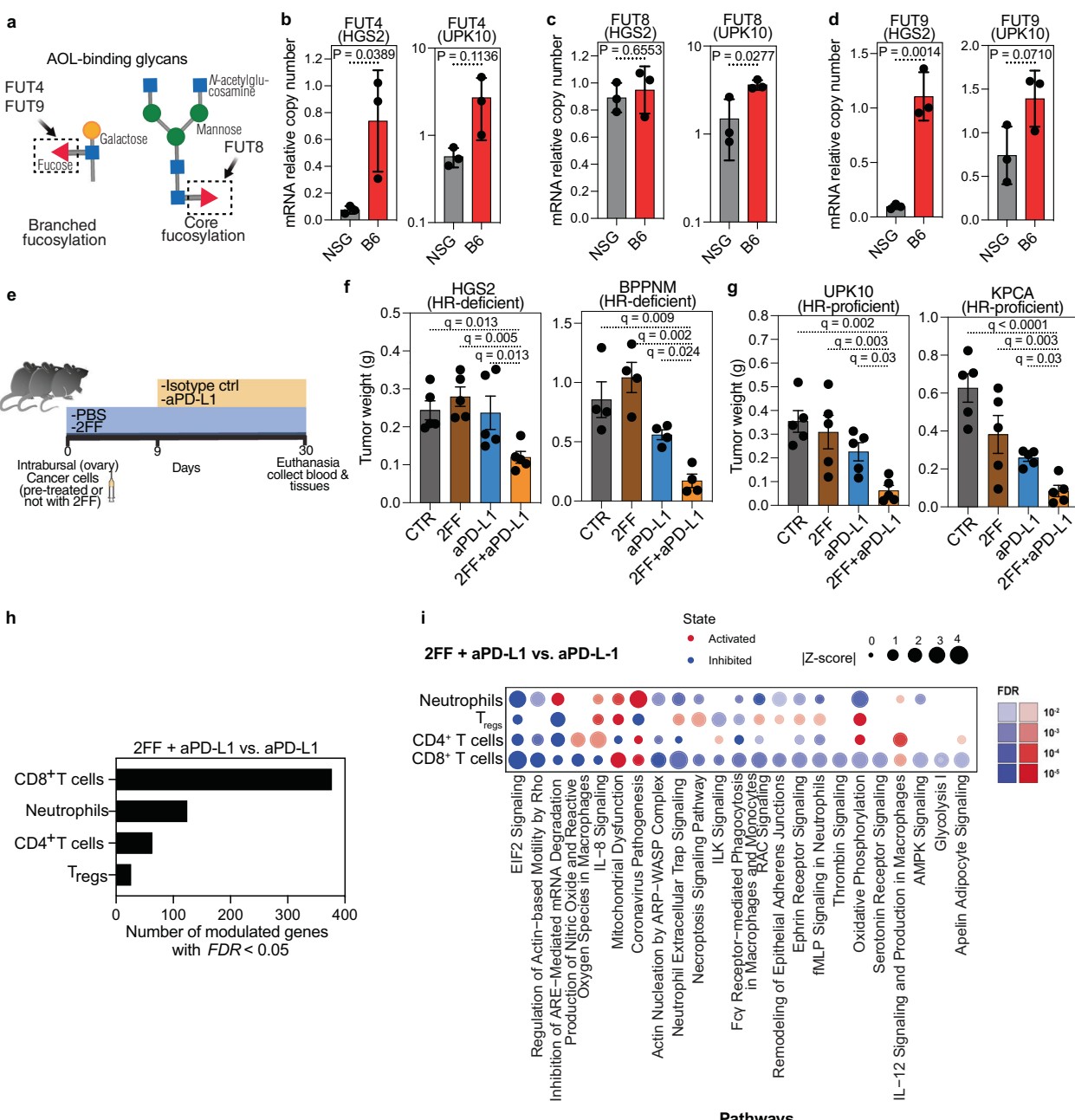

**Fig. 2 | Inhibiting fucosylation enhances sensitivity of ovarian tumors to anti-PD-L1 immunotherapy in vivo. a** Representation of AOL-binding glycans enriched in tumors grown in immunocompetent (C57BL/6; B6) compared to immunocompromised (NSG) environments. **b–d** Relative expression of genes encoding the indicated fucosyltransferases involved in tumors grown in B6 compared to NSG mice as determined by RT-qPCR. n = 3 biologically independent samples. P-values were calculated using two-tailed t-test. Error bars represent Mean with SD. **e** Schematic representation of in vivo experiments examining the impact of 2FF treatment on the response of ovarian cancers to anti-PD-L1 immunotherapy. **f-g** Weight of orthotopic tumors formed by HR-deficient HGS2 or BPPNM (**f**) or HR-proficient UPK10 or KPCA (**g**) cells in C57BL/6 mice following the indicated treatments with 2FF, anti-PD-L1 antibody or in combination (n = 5 mice per group with exception of n = 4 mice per group for BPPNM model). Two-tailed P-values were calculated by ANOVA corrected by the Benjamini, Krieger and Yekutieli method to generate q-values. Error bars represent mean with SEM. **h** Number of genes whose expression was significantly modulated in mice treated with 2FF in addition to anti-PD-L1 compared to mice treated with anti-PD-L1 alone. n = 3 77 genes for CD8+ T cells, n = 124 genes for neurophils, n = 63 for CD4+ T cells, and n = 26 for Tregs cells. **i** Ingenuity pathway analyses of pathways significantly modulated in different immune cells from mice treated with the combination of 2FF and anti-PD-L1 compared to mice treated with anti-PD-L1 alone. For single cell RNA-seq, n = 1 mouse from each of the indicated groups. Source data are provided as a Source Data file.

2DG treatment had no impact in enhancing the efficiency of anti-PD-L1 to eliminate HR-deficient BPPNM tumors (Fig. 6d).

To gain deeper insights into how 2DG treatment mediates the anti-tumor response, we conducted single-cell RNA sequencing to characterize the tumor immune microenvironment in mice treated with or without 2DG in combination with anti-PD-L1 treatment. Both 2DG and anti-PD-L1 treatments enhances the infiltration of immune cells to the tumor (Supplementary Fig. 11a, b). We observed that, compared to treatment with anti-PD-L1 alone, the combination of 2DG with anti-PD-L1 led to significant modulation of several pathways in CD8+ T cells, including immune response (PD1, PD-L1 cancer immunotherapy pathway) that is relevant to response to anti-PD-L1 treatment (Fig. 7a). Notably, the combination treatment significantly increased the expression of several molecules associated with CD8+ T

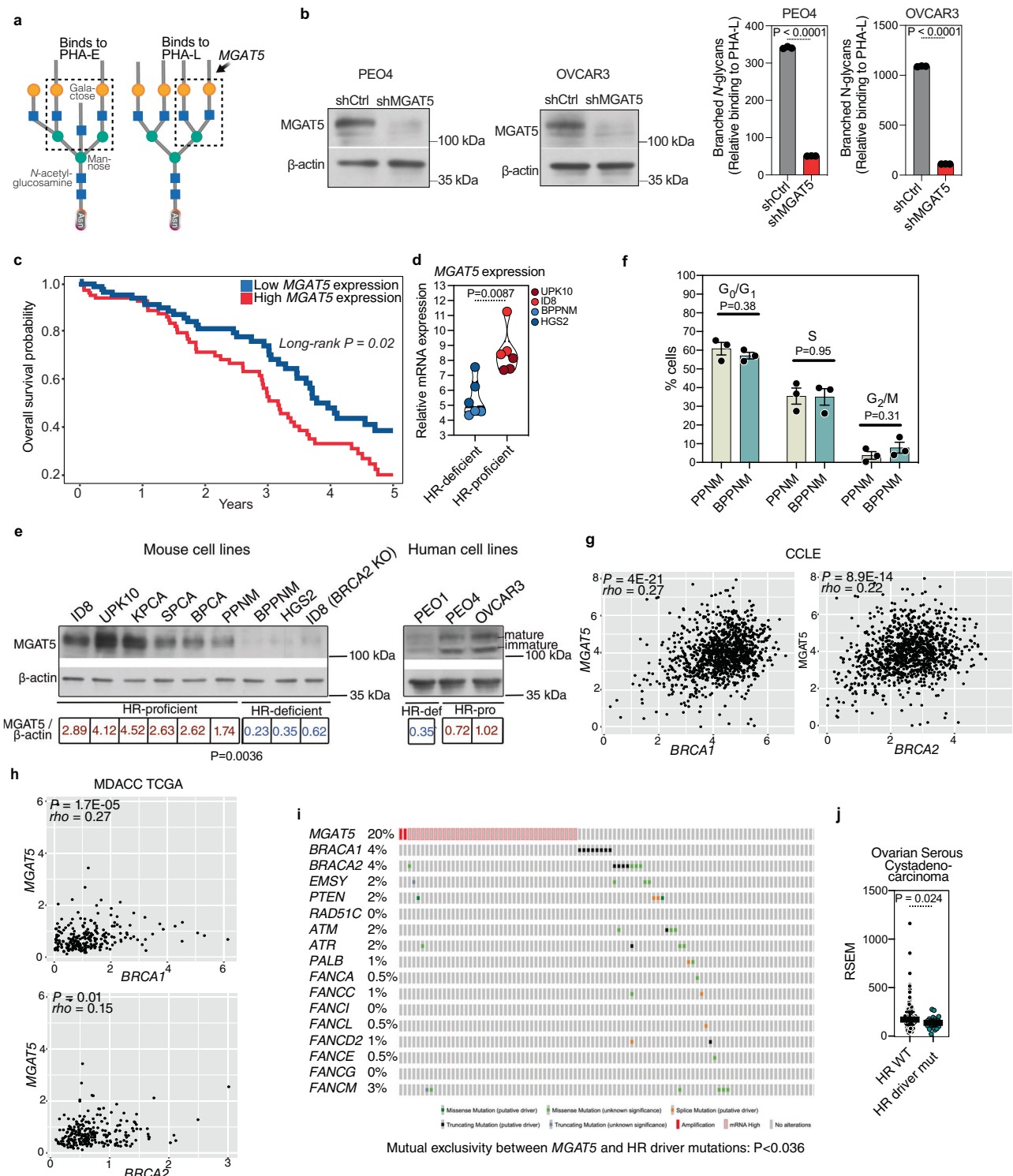

cell activation and cytotoxicity, such as IFNγ, compared to treatments with anti-PD-L1 or 2DG alone (Fig. 7b). This observation was confirmed at the protein level using flow cytometry (Fig. 7c and Supplementary Fig. 12). We also observed that the combination treatment induced higher levels of PD-1 on CD8⁺ T cells but has no effect on the levels of PD-L1 on CD45⁻ tumor cells (Supplementary Fig. 11c, d). The impact of the treatment on PD-1 expression levels could be explained by the fact that PD-1 serves as a marker of activation on T cells, further supporting the notion that blocking branched *N*-glycans enhances T cell activation in tumors, which contributes to the observed anti-tumor activity. As a control, we next

examined the effects of 2DG treatment on the anti-tumor activity of anti-PD-1 treatment. As shown in Fig. 7d, while 2DG enhanced the anti-tumor activity of both anti-PD-1 and anti-PD-L1 treatments, the effects were more pronounced with the anti-PD-L1 blockade. Considering that anti-PD-1 blockade may not fully effective in blocking all PD-1/PD-L1 interactions, 2DG could enhance the effects of both anti-PD-1 and anti-PD-L1. However, it should enhance anti-PD-L1 more than anti-PD-1, as we observed. This is because, in the case of anti-PD-1, it only affects the residual PD-1/PD-L1 binding not blocked by anti-PD-1. Whereas with anti-PD-L1, its effects should be more generalized.

**Fig. 3 | HR-proficient ovarian tumors exhibit elevated levels of MGAT5, the enzyme that catalyzes branched N-glycans, compared to HR-deficient tumors.**
**a** The glycan structures PHA-E (left) and PHA-L (right) are specific for branched *N*-glycans that are catalyzed by the MGAT5 enzyme. **b** HR-proficient cell lines PEO4 and OVCAR3 expressing shMGAT5 and shControl were validated for MGAT5 knockdown by immunoblot (left panels) and examined for the frequency of PHA-L binding to the cells (right panels). *P*-values were calculated using two-tailed *t*-test. *n* = 3 biologically independent samples. Error bars represent mean with SD. **c** Kaplan–Meier analysis of overall survival (OS) based on *MGAT5* mRNA levels in the TCGA HGSOC database. *n* = 373 patients. Quartile-grouping (low, high) was chosen in the analysis. *P*-value was calculated by Log-rank test. **d** Relative expression of *MGAT5* mRNA in HR-proficient (UPK10 and ID8) versus HR-deficient (BPPNM and HGS2) cells. *n* = 3 biologically independent samples. *P*-values were calculated using non-parametric two-tailed *t*-test. **e** Expression of MGAT5 protein in HR-proficient cell lines and HR-deficient cell lines of mouse (right) and human (left). *P*-values were calculated using two-tailed *t*-test. **f** Cell cycle distribution as determined by FACS analysis in the indicated BPPNM and PPNM cells. *n* = 3 biologically independent samples. Error bars represent mean with SEM. *P*-values were calculated using two-tailed *t*-test. **g** Positive Spearman's r correlations (two-tailed) between *MGAT5* and *BRCA1/2* expression in 1139 *BRCA1/2* wildtype cancer cell lines across cancer types in the Cancer Cell Line Encyclopedia RNAseq database. **h** Positive Spearman's r correlations (one-tailed) between *MGAT5* and *BRCA1/2* expression in 236 HGSOC tumors with wildtype *BRCA1/2* in the TCGA dataset. **i** Mutual exclusivity between *MGAT5* amplification and/or overexpression and genetic alterations in the HR pathway in the TCGA HGSOC dataset. *n* = 201 patients. *P*-value was calculated using one tailed hypergeometric test. **j** *MGAT5* is expressed at significantly higher levels in HGSOC tumors with wildtype genes of the HR pathway (*n* = 160) versus HGSOC tumors with driver mutations in HR pathway genes (*n* = 22) in the TCGA HGSOC dataset. One-tailed *P*-value was calculated by non-parametric *t*-test. Error bars represent median with IQR. Source data are provided as a Source Data file.

To conclusively demonstrate that the observed increase in sensitivity to anti-PD-L1 immunotherapy induced by 2DG combination is driven by its effect on CD8+ T cells, we conducted an in vivo experiment where we depleted CD8+ T cells using an anti-CD8 antibody in HR-proficient tumor-bearing mice and then treated them with or without the combination of 2DG and anti-PD-L1. The results showed that the suppression of tumor growth (Fig. 7e) and the improvement in the survival of tumor-bearing mice by the combination treatment were significantly abolished when CD8+ T cells were depleted (Fig. 7f). These findings support that inhibiting branched *N*-glycans enhances the sensitivity of HR-proficient tumors to immunotherapy by modulating CD8+ T cells.

## Discussion

In this study, we profiled glycomic alterations on EOC cells that drives immune evasion. Our findings unveiled several glycomic alterations, such as fucosylation, that are employed by EOC cells to broadly evade cellular immune responses. Notably, we also discovered branched *N*-glycans as a means to evade the CD8+ T cell response by HR-proficient EOCs, which represent a vulnerability that can be therapeutically explored with precision. Thus, these results offer valuable mechanistic insights into our understanding of the mechanisms by which EOC evades immune surveillance in the tumor microenvironment. Consequently, these discoveries hold the potential to guide the development of therapeutic strategies aimed at boosting anti-tumor immunity with precision by targeting HR-proficient EOCs, a critical unmet need in clinical management of this devasting disease.

Our study revealed a significant elevation of fucosylated glycans on ovarian cancers when subjected to immune pressures. Consistently, branched fucosylated glycoantigens, like Lewis antigens, interact with the C-type lectin DC-SIGN, which is expressed on various immune cells. This raises the possibility that tumor cells may engage with DC-SIGN to modulate immune cell functions[13]. In addition, core fucosylation has been associated with the progression of several cancers, including melanoma and breast cancers[15,42–47]. Moreover, it has been implicated in influencing PD-1/PD-L1-mediated immune evasion[14,30]. Our findings corroborate these existing associations by extending them into EOCs. Notably, EOC cells utilize fucosylation regardless of HR-status. This suggests that targeting fucosylation is a broadly applicable strategy to sensitize EOCs to ICB. Further investigation is warranted to understand the specific mechanisms through which immune pressure increases fucosylation on EOC cells and, conversely, how fucosylation promotes immune evasion and progression of EOC.

Our study highlighted a crucial glycomic difference between HR-proficient and -deficient EOCs, namely branched *N*-glycans. Elevated branched *N*-glycans can influence various cellular processes and immunological functions central to cancer progression, including cell growth, invasion, metastasis, metabolism, stemness maintenance, and immune surveillance. Consistent with this, high levels of MGAT5 expression, responsible for the synthesis of branched *N*-glycans, have been linked to poor prognosis in various cancer types[48,49]. Our identification of elevated MGAT5 and subsequent branched *N*-glycans in HR-proficient tumors not only shed light on the upstream mechanism driving this upregulation but also revealed its downstream impact on the immune system. Specifically, these branched *N*-glycans reinforce PD-1/PD-L1 binding, thereby reducing the effectiveness of anti-PD-L1 immunotherapy. This discovery allowed us to explore a potential avenue to enhance the sensitivity of HR-proficient cancers to anti-PD-L1 immunotherapy, namely by inhibiting these branched *N*-glycans. Considering the pressing clinical need to find effective treatments for HR-proficient EOCs, our findings represent a potential therapeutic strategy to sensitize these tumors to ICB such as anti-PD-L1 treatment. Moreover, given the fact that fucosylation is also implicated in immune response to HR-proficient EOC cells, our findings suggest that tumor cells might employ distinct yet complementary strategies to evade immune surveillance. This insight highlighted the possibility to develop personalized approaches by leveraging unique glycomic changes in the diverse subtypes of ovarian cancers.

In summary, these results underscore the potential of targeting alterations in glycans as an effective therapeutic strategy to restore anti-tumor immunity. Notably, our findings identify branched *N*-glycans as promising therapeutic targets whose inhibition sensitizes HR-proficient tumors to ICB by overcoming immune evasion. While this study focused on branched *N*-glycosylation and their roles in modulating immune responses against HR-proficient EOCs, our assessment of the glycomes of EOC with or without immune pressures, as well as in comparing HR-proficient and deficient EOCs, identified several glycomic changes that warrant further investigation. Investigating these changes can open avenues to enhance the efficacy of immunotherapies against EOCs. A limitation of our study is that the single cell RNA-seq analysis was based on one mouse from each group. Nonetheless, we validated key findings by functional studies such as depletion of CD8+ T cells. Together, our study not only contributes to a deeper understanding of the glycomic alterations associated with ovarian cancers but also provides critical insights into the development of urgently needed therapeutic strategies by leveraging these insights.

## Methods
### Animal models
Only female mice were used in this study because the current study focuses on ovarian cancer. The study protocols were approved by the Institutional Animal Care and Use Committee at The Wistar Institute (protocol number: 2021205) or The University of Texas MD Anderson Cancer Center (protocol number:00002384-RN00). All animal experiments were carried out in accordance with the relevant guidelines. Mice were maintained at 22–23 °C with 40–60% humidity and a

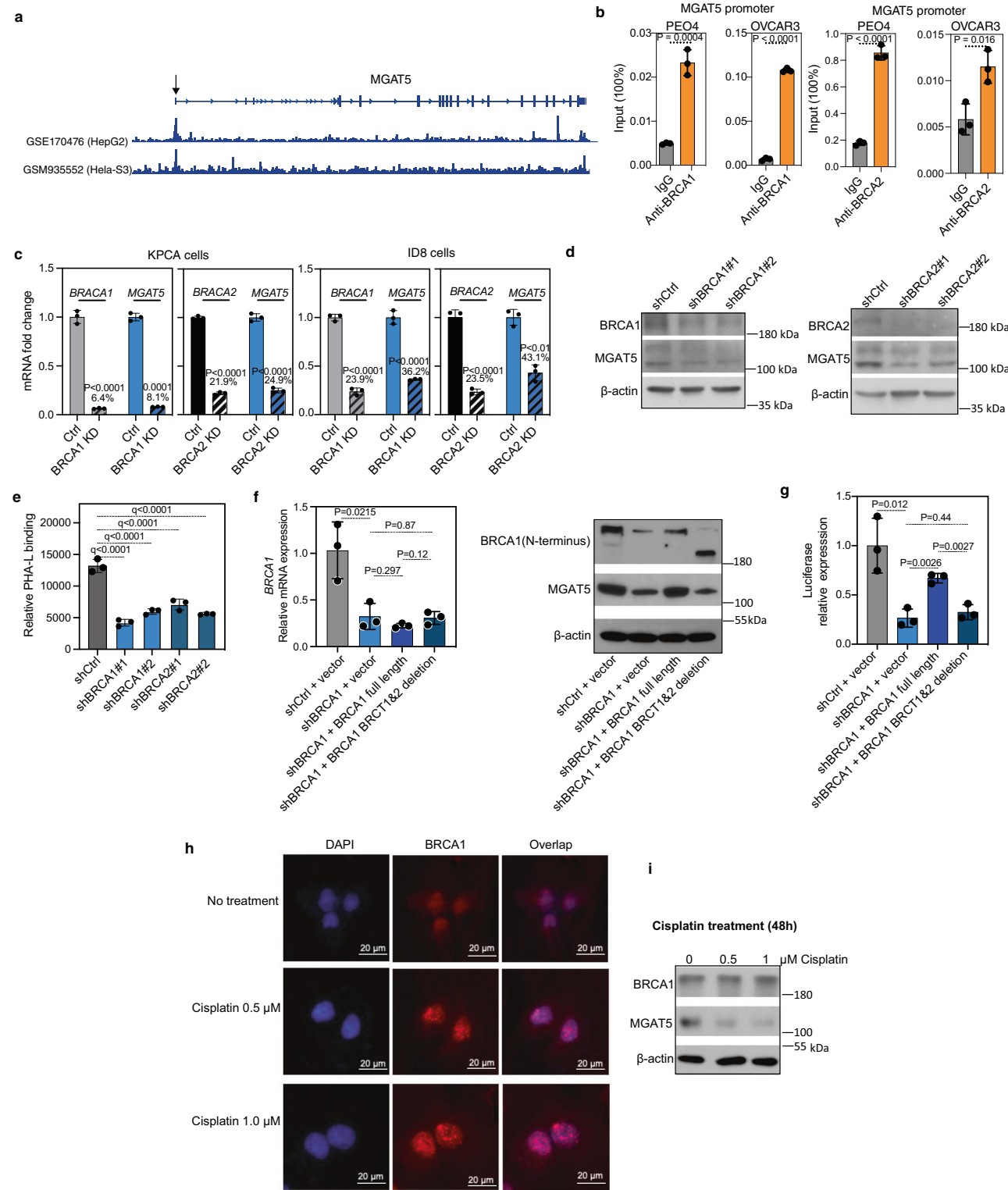

12 h light–12-h dark cycle. The maximum tumor burden (10% of the body weight as determined by The Wistar Institute and The University of Texas MD Anderson Cancer Center IACUC guidelines) has not been reached. For mouse models, 2 different strains were used in this study. 6–8 weeks old female C57BL/6 mice (Strain Code: 027, RRID:MGI:2159769) from Charles River Laboratories, and 6–8 weeks old female NSG mice (NOD.Cg-Prkdcscid Il2rgtm1Wjl/SzJ, RRID: IMSR JAX:005557) from Jackson Laboratory were housed and maintained in individual microisolator cages in a rack system capable of managing air exchange with filters.

## Cell lines and culture conditions

Only female cancer cell lines were used because the current study focuses on ovarian cancer. Human high grade serous ovarian cancer (HGSOC) cell lines PEO1 (Fox Chase Cancer center, RRID:CVCL_2686), PEO4 (Sigma-Aldrich, Cat#: 10032309, RRID:CVCL_2690) and OVCAR3 (ATCC, RRID:CVCL_0465) were cultured in RPMI-1640 with 10% FBS and 1% penicillin/streptomycin at 37 °C supplied with 5% $CO_2$. Human embryonic kidney cell line HEK-293T (ATCC, RRID:CVCL_0063) was cultured in DMEM with 10% FBS and 1% penicillin/streptomycin at 37 °C supplied with 5% $CO_2$.

**Fig. 4 | BRCA1/2 transcriptionally regulate MGAT5 expression. a** BRCA1 ChIP-seq tracks in the *MGAT5* gene locus in the indicated HepG2 and Hela-S3 cells in the ENCODE database. **b** The association of BRCA1 (left) and BRCA2 (right) with the *MGAT5* gene promoter in the indicated cells was examined by ChIP–qPCR analysis. An isotype-matched IgG was used as a negative control. *P*-values were calculated using two-tailed *t*-test. *n* = 3 biologically independent samples. Error bars represent mean with SD. **c** HR-proficient cell lines KPCA and ID8 expressing shBRCA1, shBRCA2 or shControl were validated for BRCA1/2 knockdown and examined for *MGAT5* expression by qRT-PCR. *P*-values were calculated using two-tailed *t*-test. *n* = 3 biologically independent samples. Error bars represent mean with SD. **d** HR-proficient cell line OVCAR3 expressing shBRCA1, shBRCA2 or shControl were validated for BRCA1/2 knockdown and examined for MGAT5 expression by immunoblot. The experiment was repeated independently for 3 times with similar results. **e** Levels of PHA-L binding to OVCAR3 expressing shBRCA1, shBRCA2 or shControl. Two-tailed *P*-values (*q*-values) were calculated using ANOVA tests, corrected using the Benjamini and Hochberg method. *n* = 3 biologically independent

samples. Error bars represent mean with SD. **f** Validation endogenous BRCA1 knockdown by RT-qPCR using primers targeting the 3′ UTR region of the BRCA1 gene as well as ectopic expression of the indicated full length or truncated BRCA1 by immunoblot using an antibody against the N-terminus of BRCA1. *P*-values were calculated using two-tailed *t*-test. *n* = 3 biologically independent samples. Error bars represent mean with SD. **g** *MGAT5* gene promoter luciferase activity in control and BRCA1 knockdown OVCAR3 cells with full length BRCA1 and truncated BRCA1 re-expression. *P*-values were calculated using two-tailed *t*-test. *n* = 3 biologically independent samples. Error bars represent mean with SD. **h** HR-proficient cell line OVCAR3 treated with 0.5 μM and 1 μM cisplatin (Selleckchem, Cat#: S1166) for 48 h were examined for BRCA1 foci by immunofluorescence. The experiment was repeated independently for 3 times with similar results. **i** HR-proficient cell line OVCAR3 treated with 0.5 μM and 1 μM cisplatin (Selleckchem, Cat#: S1166) for 48 h were examined for MGAT5 expression by immunoblot. The experiment was repeated independently for 3 times with similar results. Source data are provided as a Source Data file.

The sources and genotypes of mouse HGSOC cell lines have been listed in Supplementary Table 1. Mouse HGSOC cell lines KPCA, BPCA, SPCA, PPNM, BPPNM, HGS2 (RRID:CVCL_B5GW), ID8 (RRID:CVCL_IU14) and UPK10 were cultured in fallopian tube cells media (FT-media); DMEM supplemented with 1% insulin–transferrin–selenium (Thermo Fisher Scientific; ITS-G, 41400045), EGF (Sigma-Aldrich; E4127, 2 ng/mL), 4% heat-inactivated fetal bovine serum (Thermo Fisher Scientific; IFS, F4135), and 1% penicillin/streptomycin at 37 °C supplied with 5% $CO_2$.

All the cell lines were authenticated using short tandem repeat DNA profiling. Mycoplasma was tested monthly using mycoplasma PCR detection kit (Sigma-Aldrich, Cat#: MP0035).

## Plasmids and lentivirus infection

pLKO.1-human MGAT5-shRNA1 (TRCN0000036060), pLKO.1-human MGAT5-shRNA2 (TRCN0000036063), pLKO.1-human BRCA1-shRNA1 (TRCN0000039833), pLKO.1-human BRCA1-shRNA2 (TRCN0000039837), pLKO.1-human BRCA2-shRNA1 (TRCN0000009825), pLKO.1-BRCA2-shRNA2 (TRCN0000010306), pLKO.1-mouse BRCA1-shRNA (TRCN0000042561), pLKO.1-mouse BRCA2-shRNA (TRCN0000071009), pLKO.1-mouse MGAT5-shRNA (TRCN0000018755), human CD19 (hCD19) ORF and human BRCA1 ORF were obtained from the Wistar Institute Molecular Screening and Protein Expression Facility. pLKO.1-scramble shRNA is obtained from Addgene (#1864, RRID: Addgene_1864). pGIPZ-human MGAT3 shRNA (CCGACGACGTCTT-CATCAT) and pGIPZ non-silencing control were obtained from MD Anderson Cancer Center Functional Genomics Core. To construct pLVX-hCD19 plasmid, hCD19 ORF was PCR-amplified and cloned into pLVX-M-puro (Addgene, #125839, RRID: Addgene_125839). To construct pGL4.10-MGAT5 promoter plasmid, 2000 bp of MGAT5 promoter was PCR-amplified using OVCAR3's genomic DNA as the template and cloned into pGL4.10[luc2] (Promega, E6651). To construct pCDH-BRCA1 and pCDH-truncated BRCA1 plasmid, full length and truncated BRCA1 was PCR-amplified using BRCA1 ORF as the template and cloned into pCDH-CMV (Addgene, #72265, RRID: Addgene_72265). To generate the expression plasmid encoding anti-CD19 CAR T cells, the FMC63 mouse-derived anti-human CD19 single chain variable region[50] was synthesized by Genscript and cloned into the pTRPE lentiviral expression plasmid backbone[51]. An EF1a promoter drive the expression of a GFP reporter upstream of a T2A cleavage site, followed by the chimeric antigen receptor consisting of the FMC63 mouse anti-human single chain variable region followed by the CD8a hinge, CD28 transmembrane domain, the CD28 intracellular costimulatory domain, and the CD3z chain. HEK-293T cells were transfected by Lipofectamine 2000 for lentivirus package. Lentivirus was collected and filtered with 0.45-mm filter 48 h after transfection. Cells infected with lentivirus were selected in 1 mg/ml puromycin or 1 mg/ml blasticidin 48 h after infection.

## Cell cycle analysis

Cell was harvested and washed twice with PBS, then fixed with 70% ethanol in PBS at 4 °C overnight. Cells were washed twice with PBS and incubated with a mix containing 100 mg/ml RNAse A (Thermo Fisher, EN0531) and 50 mg/ml propidium iodide (Thermo Fisher, P3566) for 15 min. Then the samples were examined by flow cytometry.

## Colony formation assay

Three thousand cells were seeded into 24-well tissue culture plates and cultured in control medium for 1 week. In the last 2 days, the control medium was replaced with medium containing 2DG. Afterward, colonies were stained with 0.05% crystal violet, and the signal intensity was quantified using the National Institutes of Health ImageJ software (version 1.53a).

## Antibodies and immunoblotting

Whole-cell lysate was extracted using RIPA buffer (50 mM Tris pH 8.0, 150 mM NaCl, 1% Triton X-100, 0.5% sodium deoxycholate, 1 mM EDTA, 1 mM dithiothreitol, and 1 mM PMSF) on ice. Proteins were separated by SDS-PAGE gel and transferred to a PVDF membrane (Millipore, IPVH00010). The membranes were blocked with 5% non-fat milk and incubated with primary antibodies overnight at 4 °C with gentle agitation using the following antibodies and dilutions: anti-β-actin (1:1000; Cell Signaling, Cat#: 4970, RRID:AB_2223172), anti-BRCA2 (1:500; Cell Signaling, Cat#: 10741, RRID:AB_2797730), anti-BRCA1 (c-terminus, 1:200; Santa Cruz, Cat#: sc-6954, RRID: AB_626761), anti-BRCA1 (n-terminus, 1 μg/mL, R&D Systems, Cat#: AF2210, RRID:AB_2067618), anti-MGAT5 for mouse (1:500; Thermo Fisher, Cat#: PA5-87988, RRID:AB_2804566), anti-MGAT5 for human (1:500; R&D Systems, Cat#: MAB5469, RRID:AB_10972310). Objective signals were amplified with HRP-conjugated secondary antibodies (1:2000; Cell Signaling, Cat#: 7076, RRID:AB_330924; 1:3000; Cat#: 7074, RRID:AB_2099233; and 1:5000; Thermo Fisher, Cat#: 31402, RRID:AB_228395) and detected by chemiluminescent substrate (Thermo Fisher, Cat#: 34094).

## Glycomic analysis using lectin microarrays

The protocols were approved by the Institutional Animal Care and Use Committee (IACUC) at The Wistar Institute. 1 × 10⁶ mouse ovarian cancer cells with different genotypes were unilaterally injected into the ovarian bursa sac of 6–8 week-old female immunocompetent C57BL/6 mice or immunocompromised NSG mice. Tumors were collected after 4 weeks, then tumors were chopped and digested with the Mouse Dissociation Kit (Miltenyi Biotec, 130-096-730) according to the manufacturer's instructions. Single cells were then harvested with a 70 mm strainer and used for staining. MoFlo Astrios EQ (Beckman) was used to sort out immune cells from tumor cells by BV-711 anti-mouse CD45 antibody (1:100; Biolegend, 103147,

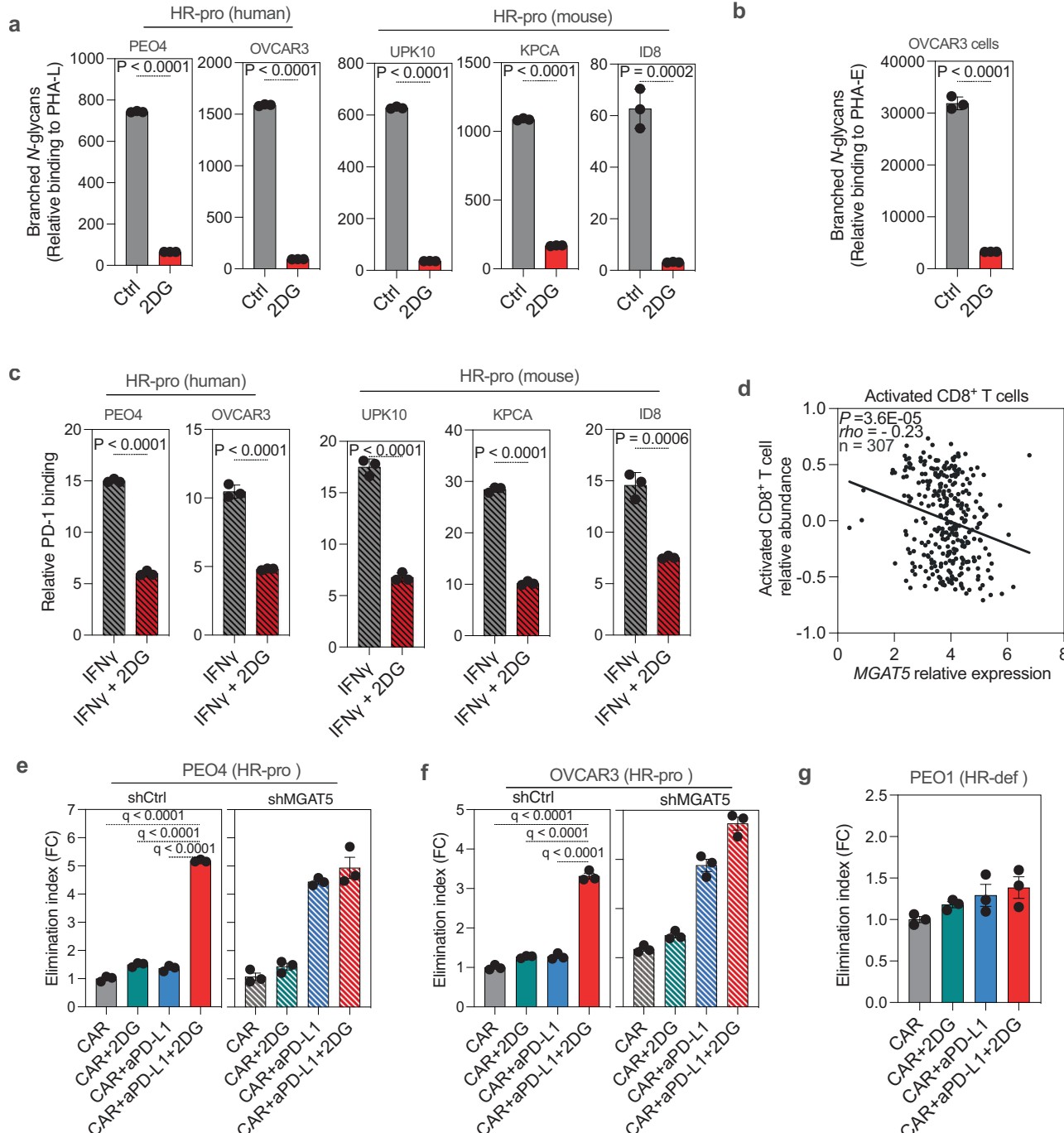

**Fig. 5 | Branched *N*-glycans increase the binding affinity of PD-L1 on HR-proficient cancer cells to PD-1 on CD8⁺ T cells. a** Frequency of PHA-L binding to HR-proficient cells treated with 1 mM 2DG for 48 h. *P*-values were calculated using two-tailed *t*-test. *n* = 3 biologically independent samples. Error bars represent mean with SD. **b** Frequency of PHA-E binding to HR-proficient cell OVCAR3 treated with 1 mM 2DG for 48 h. *P*-values were calculated using two-tailed *t*-test. *n* = 3 biologically independent samples. Error bars represent mean with SD. **c** Frequency of recombinant PD-1 binding to PD-L1⁺ HR-proficient cells treated with 1 mM 2DG for 48 h. *P*-values were calculated using two-tailed *t*-test. *n* = 3 biologically independent samples. Error bars represent mean with SD. **d** Reverse correlation between *MGAT5* mRNA expression and tumor-infiltrating activated CD8⁺ T cells in the TCGA HGSOC

dataset (two-tailed Spearman's r correlation test), data was analyzed by TISIDB. **e** Killing of HR-proficient PEO4 cells expressing CD19 was measured after coculture with anti-CD19 CAR T cells at the 1:6 E:T ratio. *n* = 3 biologically independent samples. **f** Killing of HR-proficient OVCAR3 cells expressing CD19 was measured after coculture with anti-CD19 CAR T cells at the 1:6 E:T ratio. *n* = 3 biologically independent samples. **g** Killing of HR-deficient PEO1 cells expressing CD19 was measured after coculture with anti-CD19 CAR T cells at the 1:6 E:T ratio. *n* = 3 biologically independent samples. Two-tailed *P*-values were calculated using ANOVA used for analyses and then corrected using the Benjamini, Krieger, and Yekutieli method to generate *q*-values unless otherwise stated. Error bars represent mean with SEM. Source data are provided as a Source Data file.

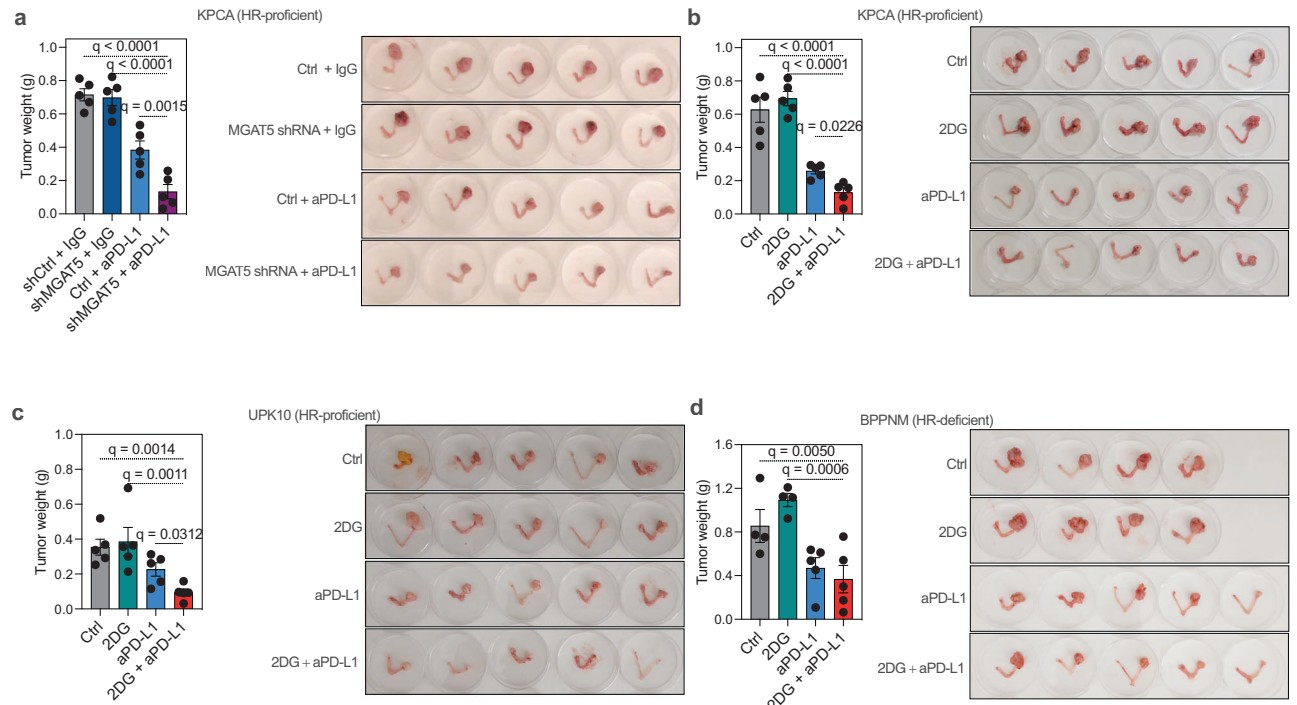

**Fig. 6 | Inhibiting branched *N*-glycans enhances the sensitivity of HR-proficient ovarian tumors to immune checkpoint blockade. a** Weight and images of orthotopic tumors formed by HR-proficient KPCA with shCtrl or shMGAT5 in C57BL/6 mice following the treatments with anti-PD-L1 antibody ($n = 5$ mice per group). Two-tailed *P*-values were calculated by ANOVA corrected by the Benjamini, Krieger and Yekutieli method to generate *q*-values. Error bars represent mean with SEM. **b–d** Weight and images of orthotopic tumors formed by HR-proficient KPCA (**b**) or UPK10 (**c**), or HR-deficient BPPNM (**d**) cells in C57BL/6 mice following the

indicated treatments with 2DG, anti-PD-L1 antibody or in combination ($n = 5$ mice per group for anti-PD-L1 and the combination groups with exception of $n = 4$ mice per group for control and 2DG groups). Please note that the control and anti-PD-L1 treatment groups were the same as those presented in Fig. 2g, h as they were simultaneously performed. Two tailed *P*-values were calculated by ANOVA corrected by the Benjamini, Krieger and Yekutieli method to generate *q*-values. Error bars represent mean with SEM. Source data are provided as a Source Data file.

RRID:AB_2564383) staining. A lectin microarray platform was used to profile 96 different glycan structures on the surface of sorted tumor cells. The lectin array employs a representative panel of immobilized lectins with known glycan structure binding specificity. Cell-membrane proteins were labeled with Cy3 dye (Sigma-Aldrich, Cat# GEPA23001) and hybridized to the lectin microarray. The resulting lectin chips were scanned for fluorescence intensity on each lectin-coated spot using an evanescent-field fluorescence scanner (Rexxam Co., Ltd.). All samples were run in triplicate, and the average of the triplicate was used for analysis. Data were normalized using the global normalization method.

### Reverse transcription and quantitative Real-Time PCR (qRT-PCR)

Total RNA was extracted using Trizol reagents (Thermo Fisher, Cat#: 15596026) according to the manufacturer's protocol. RNA was reverse transcribed with the High-Capacity cDNA Reverse Transcription kit (Thermo Fisher, Cat#: 4368813). qRT-PCR was performed using the QuantStudio 3 Real-Time PCR System (Thermo Fisher). Following primers for PCR were purchased from Integrated DNA Technologies; human MGAT3 (forward: 5'-ACGTCAACCACGAGTTCGACCT-3' and reverse: 5'-AAGCCGTGAAGTTGGACTCGCA-3'), human BRCA1 UTR (forward: 5'- GATCCCACCAGGAAGGAAGC-3' and reverse: 5'-AGTCTT-CACTGCCCTTGCAC-3'), human B2M (forward: 5'-CCACTGAAAAA-GATGAGTATGCCT-3' and reverse: CCAATCCAAATGCGGCATCTTCA-3'), mouse *Fut4* (forward: 5'- CGCGAATGGATGTGCTTTCCTG -3' and reverse: 5'-CGCTTTCCCGAGATTTACCC-3'), mouse *Fut8* (forward: 5'-GAGCACAGATGGAGACAGGGAA-3' and reverse: 5'-TCACTCTGCGAG-CAGTCTTCAG-3'), mouse *Fut9* (forward: 5'-CGCGAATGGATGTG CTTTCCTG-3' and reverse: 5'-GACTTCTGCGTAAGGATGCTGG-3'),

mouse *Brca1* (forward: 5'-CGAGGAAATGGCAACTTGCCTAG-3' and reverse: 5'-TCACTCTGCGAGCAGTCTTCAG-3'), mouse *Brca2* (forward: 5'-GAGCACAGATGGAGACAGGGAA-3' and reverse: 5'-CAGACGGAGA TGTCGCCTTTCT-3') and mouse b-Actin (forward: 5'-CTGGACCCAG-GAAGATGAAAG-3' and reverse: 5'-GCCAAACCAGCACCATTTAC-3').

### Immunofluorescence

Cells were fixed with 4% paraformaldehyde (PFA) and permeabilized with 1% Triton-X in PBS. Primary antibody anti-BRCA1 (1:50; Santa Cruz, Cat#: sc-6954, RRID: AB_626761) was incubated overnight at 4 degrees in 5% goat serum. Alexa Fluor™ 568 conjugated secondary antibody (2 μg/mL; Thermo Fisher, Cat# A-11004) were incubated for 1 h at room temperature.

### Reporter assay

OVCAR3 cells ($1 \times 10^5$ per well) were seeded in 12-well plate and co-transfected with 180 ng pGL4.10-MGAT5 promoter plasmid and 20 ng pRL-SV40 (Promega, Cat#: E223A) using Lipofectamine 2000 (Invitrogen). Fourty-eight hours later, a Dual-Luciferase Reporter Assay System (Promega, Cat#: E1910) was used for the luminescence assay. Luminescence was measured using a BioTek Synergy H1 Microplate Reader (Agilent).

### Immune cell analysis

Tumors were chopped and digested with Mouse Dissociation Kit (Miltenyi Biotec, 130-096-730) according to the manufacturer's instructions. Single cells were then harvested with 70 mm strainer and used for staining. Live/dead cells were discriminated by viability staining Kit (Thermo Fisher, Cat#: L34965). Fc blocking (BD Biosciences, Cat#: 553142) was followed by cell surface staining in FACS

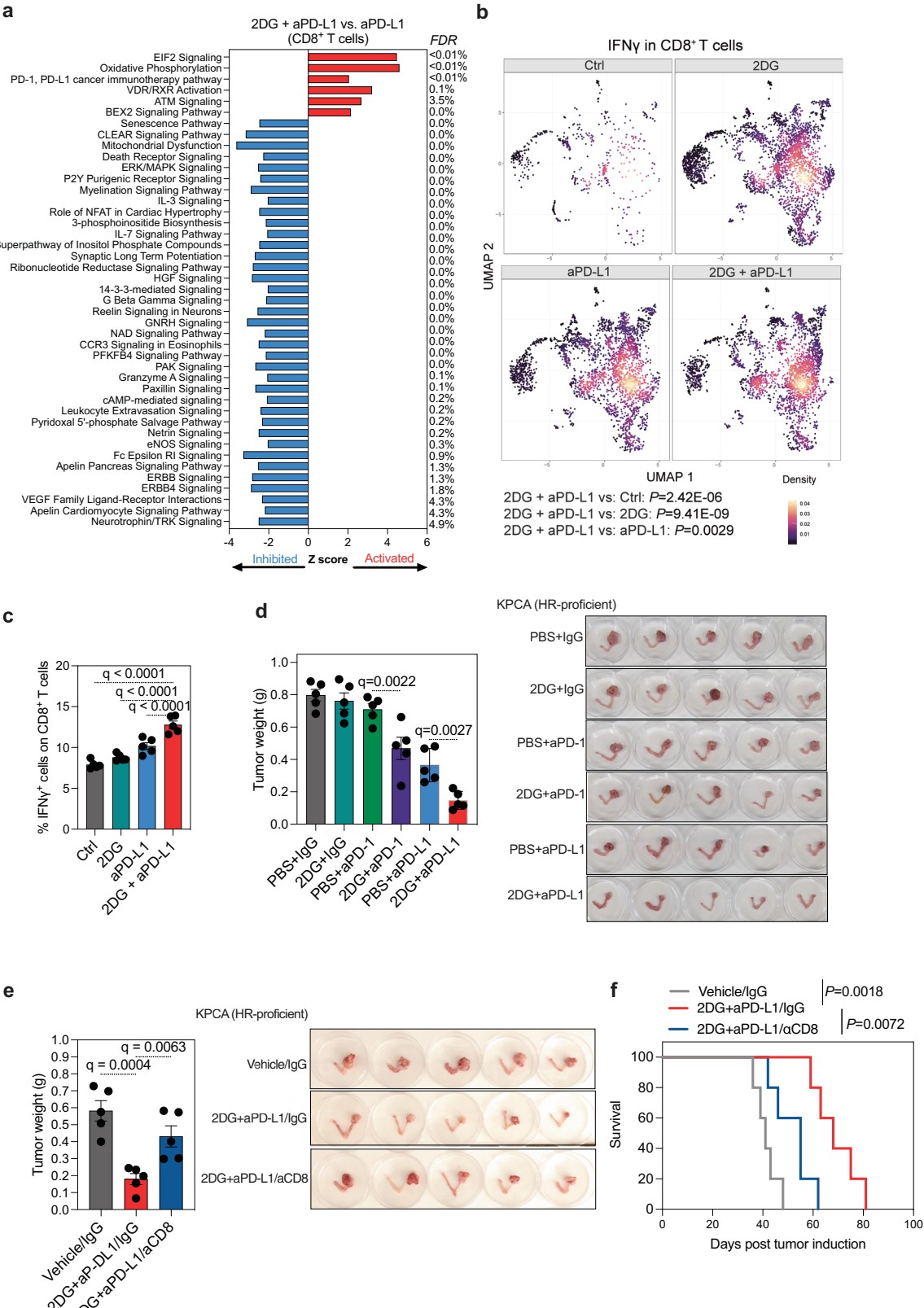

buffer (2% FBS in PBS buffer) using antibodies against CD3ε (1:100; Biolegend, Cat#: 100320, RRID:AB_312684), CD45 (1:100; Biolegend, Cat#: 103147, RRID:AB_2564383), CD8a (1:100; Biolegend, Cat#: 100707, RRID:AB_312747), PD-1 (1:100; Biolegend, Cat#: 135209, RRID:AB_2251944) and PD-L1 (1:100; Biolegend, Cat#: 124315, RRID:AB_10897097). Intracellular staining was carried out using True-

Nuclear™ Transcription Factor Buffer Set (Biolegend, Cat#: 424401) according to manufacturer instructions. Briefly cells were fixed for 45 min at room temperature in the dark. Following fixation, cells were washed 1x in permeabilization buffer and then incubated with anti-IFNγ antibody (1:100; Biolegend, Cat#: 505840, RRID:AB_2734493). Data was acquired using flow cytometry (BD Biosciences,

**Fig. 7 | Combination treatment of 2DG and anti-PD-L1 suppressed ovarian tumors by promoting CD8+ T cell activity. a** Signaling pathways that are enriched in the combination treatment group compared to the anti-PD-L1 treatment group of KPCA tumors based on single cell RNA-seq analysis. For single cell RNA-seq, $n = 1$ mouse from each of the indicated groups. **b** IFNγ expression of CD8+ T cells in each group of KPCA tumors based on single cell RNA-seq analysis. *P*-values were calculated using two-tailed *t*-test. For single cell RNA-seq, $n = 1$ mouse from each of the indicated groups. **c** FACS analysis of IFNγ expressed CD8+ T cells in each group of KPCA tumors. $n = 5$ mice. Two-tailed *P*-values were calculated by ANOVA corrected by the Benjamini, Krieger and Yekutieli method to generate *q*-values. Error bars represent mean with SEM. **d** Inhibiting branched *N*-glycans sensitizes HR-proficient ovarian tumors to anti-PD-1 and anti-PD-L1 immunotherapies. weight and images of orthotopic tumors formed by HR-proficient KPCA cells in C57BL/6 mice following the indicated treatments with 2DG, anti-PD-1 antibody, anti-PD-L1 antibody or in combination. $n = 5$ mice. Two-tailed *P*-values were calculated by ANOVA corrected by the Benjamini, Krieger and Yekutieli method to generate *q*-values. Error bars represent mean with SEM. **e, f** Same as (Fig. 6b), but the mice were randomized into three indicated treatment groups. After completing treatment, tumors were harvested and weighed (Two-tailed *P*-values were calculated by ANOVA corrected by the Benjamini, Krieger and Yekutieli method) (**e**) or mice were followed for survival and the Kaplan–Meier survival curves for each of the indicated groups are shown (**f**) ($n = 5$ mice/group) (*P*-values were calculated by Log-rank test). Error bars represent mean with SEM. Source data are provided as a Source Data file.

LSRFortessa™ Cell Analyzer) and analyzed using FlowJo software (version 10.0).

### Chromatin immunoprecipitation (ChIP)

Cells were crosslinked with 1% formaldehyde for 10 min and then quenched by 0.125 M glycine for 5 min at room temperature. Fixed cells were lysed with ChIP lysis buffer 1 (50 mM HEPES-KOH pH 7.5, 140 mM NaCl, 1 mM EDTA pH 8.0, 1% Triton X-100, and 0.1% DOC) on ice and lysis buffer 2 (10 mM Tris pH 8.0, 200 mM NaCl, 1 mM EDTA, and 0.5 mM EGTA) at room temperature. Chromatin was digested with micrococcal nuclease (MNase Cell Signaling, Cat#: 10010) in digestion buffer (10 mM Tris pH 8.0, 1 mM CaCl2, and 0.2% Triton X-100) at 37 °C for 15 min. Nucleus products were broken down by Bioruptor pulse at high frequency. The following antibodies were used for ChIP: anti-BRCA1 antibody (Santa Cruz, sc-6954, RRID:AB_626761; 5 µg/IP) and anti-BRCA2 antibody (Cell Signaling, 10741, RRID:AB_2797730; 10 µL/IP). ChIP DNA was purified by the ChIP DNA clean and concentrator kit (Zymo Research, Cat#: D5205) and analyzed by qPCR. Primers targeting the *MGAT5* promoter were used for ChIP–qPCR are purchased from Integrated DNA Technologies; human *MGAT5* promoter forward (5″-GTCTCCTCTGACTTCAACAGCG -3″) and human *MGAT5* promoter Reverse (5″- ACCACCCTGTTGCTGTAGCCAA -3″).

### In vitro T cell killing assay

The elimination index was determined according to a previous study. Briefly, CAR T cells were co-cultured with target cells at an E:T ratio of 1:6 in RPMI 1640 fully supplemented in the absence of cytokines. After 48 h, cells were stained with propidium Iodide (Thermo Fisher, P3566), and live/dead cells were analyzed with flow cytometry (BD Biosciences, LSRII). The elimination index was calculated as follows: 1 - (ratio of live target cells with CAR T cells / ratio of live target cells in the control group).

### PD-L1/PD-1 binding assay

For the binding assay in human and mouse cancer cells, cells were first stimulated with 100 ng/ml of human recombinant IFN-gamma (Stemcell, 78020.1) or mouse recombinant IFN-gamma (Stemcell, 78021.1) for 48 h. Then, cells were harvested and incubated with recombinant human PD-1 biotinylated protein (R&D Systems, BT1086-050) or recombinant mouse PD-1 biotinylated protein (BPS Bioscience, 71118-1) for 30 min at room temperature, followed by streptavidin AF-647 (Thermo Fisher, S21374).

### Orthotopic EOC mouse models and treatments

The protocols were approved by the Institutional Animal Care and Use Committee (IACUC) at The Wistar Institute. For in vivo experiments, a sample size of at least five mice per group was determined based on the data obtained from in vitro experiments. For orthotopic xenograft models, $1 \times 10^6$ mouse ovarian cancer cells were unilaterally injected into the ovarian bursa sac of 6–8 week-old female immunocompetent C57BL/6 mice. In the 2FF treatment group, tumor cells were pre-treated with 300 µM 2FF for 3 days before injection. After tumor implantation, 2FF was administered by oral gavage at a dose of 17.5 mg/kg in PBS every 2 days. In the 2DG treatment group, once the tumors were palpable (9 days after implantation), mice were injected intraperitoneally with 2DG (500 mg/kg in PBS) every two days. After day 9, anti-mouse PD-L1 antibody (BioXcell, BE0101, RRID:AB_10949073) or anti-mouse PD-1 antibody (BioXcell, BE0273, RRID:AB_ 2687796) was intraperitoneally injected twice a week for 3 weeks at a dose of 5 mg/kg. The relevant solvent and control rat IgG antibody (BioXCell, BE0090, RRID:AB_1107780) were administered to control animals. Please note that we used the same control and anti-PD-L1 treatment groups for comparison with the combination treatment group for both 2DG and 2FF. Mice were then sacrificed on day 30, and tumor burden was examined using tumor weight as a surrogate in each treatment group. For CD8+ T cell depletion, an anti-CD8 antibody (BioXCell, Cat#: BE0117, RRID:AB_10950145, 5 mg/kg, twice per week) was used to deplete CD8 + T cells for 3 weeks during the combination treatment. An isotype-matched IgG (Bio X Cell, Cat#: BE0090, RRID:AB_1107780) was used as a negative control. Mice were then sacrificed, and tumor burden was examined using tumor weight as a surrogate in each treatment group. For survival experiments, the guidelines of The Wistar Institute IACUC were used for endpoint assessment, such as when tumor burden exceeded 10% of body weight.

### Single-cell RNA sequencing (scRNA-seq) analysis

Pre-processing of the scRNA-seq data was performed using Cell Ranger Suite (pipeline v7.0.0, https://support.10xgenomics.com) with refdata-gex-mm10-2020-A transcriptome as a reference to map reads on the mouse genome (mm10) using STAR. Low-quality cells with <250 genes with reads and cells with over 5% mitochondrial content were filtered out. Batch effect was not observed and hence not corrected for. Cell clustering, marker identification, and visualization were performed using Seurat v4. The R package SingleR was used to determine cell types of the clusters using the ImmGen dataset as a reference for cell-specific gene signatures. Cell-typing was also verified using gene markers unique to clusters. The T cell cluster was subset and reclustered to resolve the cells into subtypes. Differential expression between samples in specific clusters was performed using Wilcoxon Rank Sum Test. Statistically significant differentially expressed genes were used as inputs for enrichment analysis using Qiagen Ingenuity pathway analysis (IPA).

### Metabolism experiments

The extracellular acidification rate (ECAR) was determined following the instructions of the Agilent Seahorse XF Glycolysis Stress Test Kit (Agilent, 103020-100). Briefly, cells were pretreated with appropriate doses of 2DG for 24 h. Then, the cells (OVCAR3 cells, 20,000 per well) were seeded and incubated with 2DG in an XF96 Cell Culture Microplate the day before running the assay. The media were exchanged to XF media 1 h before the assay. Glucose, oligomycin, and

2-deoxyglucose (2-DG) were diluted into XF media and loaded into the cartridge to achieve final concentrations of 10, 2, and 50 mM, respectively.

## Flow cytometry analysis

Cells were washed with PBS containing 0.5% (w/v) BSA (PBS/BSA). All staining was carried out in this buffer. For directly labeled antibodies and lectins, cells were then washed two times with cold PBS/BSA buffer, resuspended in this buffer, and analyzed by flow cytometry. For indirect detections, cells were washed once with PBS/BSA before the addition of the appropriate secondary detection agent. Cells were then kept on ice for 20 min, washed two times with buffer, and resuspended for flow cytometry. For PD-L1 detection in mouse cancer cells, APC anti-mouse CD274 (B7-H1, PD-L1) antibody (Biolegend, 124311, RRID:AB_10612935) was used at a 1:100 dilution. For PD-L1 detection in human cancer cells, PE anti-human CD274 (B7-H1, PD-L1) antibody (Biolegend, 329706, RRID:AB_940368) was used at a 1:20 dilution. For CD19 detection, APC anti-human CD19 antibody (Biolegend, 302212, RRID:AB_314242) was used at a 1:20 dilution. For branched N-glycan in human cancer cells, 2 μg/ml fluorescein-labeled PHA-L (Vectorlabs, FL-1111-2) was used. For bisecting GlcNAc in human cancer cells, 5 μg/ml fluorescein-labeled PHA-E (Vectorlabs, FL-1121-2) was used. For branched N-glycan in mouse cancer cells, 2 μg/ml biotinylated PHA-L (Vectorlabs, B-1115-2) was used, followed by streptavidin AF-647 (Thermo Fisher, S21374).

## Statistical analysis

Statistical analyses were performed using GraphPad Prism software (version 8.0). Two-tailed Student's $t$-tests or and ANOVA corrected by the Benjamini, Krieger and Yekutieli method was used to identify significant differences. Spearman correlation analysis was used to examine the correlation between two factors. Log-rank test was used to compare the survival distributions among experimental groups. Experiments were repeated at least twice. Quantitative data are expressed as mean ± SD unless otherwise stated. No statistical method was used to predetermine sample size. No data were excluded from the analyses. All analyses were performed blindly but not randomly. All mice for animal experiments were randomized.

## Reporting summary

Further information on research design is available in the Nature Portfolio Reporting Summary linked to this article.

## Data availability

The scRNA-seq data generated in this study have been deposited in the NCBI database under accession code GSE244012. The online ChIP-seq data that we reanalyzed here are available in the Gene Expression Omnibus (GEO) database under accession codes GSE170476 and GSM935552. Mutation status, gene expression and survival data of HGSOCs were downloaded from MDACC TCGA data portal (https://bioinformatics.mdanderson.org/MQA/) and cBioPortal (https://www.cbioportal.org/). Mutation status and gene expression for cell lines were downloaded from DepMap portal (https://depmap.org/portal/download, dataset DepMap Public 22Q2). The remaining data are available within the Article, Supplementary Information or Source Data file. Source data are provided with this paper.

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

## Acknowledgements

We thank Drs. Robert A Weinberg, Ronny Drapkin and Iain McNeish for sharing mouse ovarian cancer cell lines and Dr. Junjie Chen for critical reading of the manuscript. This work was supported by US National Institutes of Health grants (R01CA202919, R01CA239128, R01CA243142, R01CA260661, R01CA276569 and P50CA281701 to R.Z.; and R01AA029859, R01AG062383, R01NS117458, R01AI165079, R01DK123733, R21AI170166, P30AI045008, and 1UM1Al126620 to M.A.M.), US Department of Defense (OC190181 and OC210124 to R.Z.), Cancer Prevention and Research Institute of Texas (Scholar in Cancer Research RR230005 to R.Z.). Support of Core Facilities was provided by Cancer Centre Support Grant (CCSG) CA010815 to The Wistar Institute and P30CA016672 to University of Texas MD Anderson Cancer Center.

## Author contributions

H.N., P.S., T.M., L.L., R.J.Z., H.L., W.Z., C.W., B.M. and M.T. performed the experiments and analyzed data. H.N., N.Z., M.A-M. and R.Z. designed the experiments. Y.Q., T.K. and A.K. performed statistical analysis. T.Y., D.T.C. and H.T. contributed key experimental materials. M.A-M. and R.Z. supervised studies. H.N, M.A-M. and R.Z. wrote the manuscript. M.A-M. and R.Z. conceived the study.

## Competing interests

The authors declare no competing interests.
