## [Peer Review File · Nature Communications]

Targeting Branched N-Glycans and Fucosylation Sensitizes Ovarian Tumors to Immune Checkpoint BlockadeREVIEWER COMMENTS

Reviewer #1 (Remarks to the Author): with expertise in glycosylation, cancer immunology

In this interesting piece of work the authors investigate how N-glycan branching specifically in HR-proficient ovarian cancer augments resistance to anti-PD-L1 therapy. Using KO studies and inhibition with 2-DG this resistance could be reverted and sensitize ovarian cancer cells to checkpoint blockade.

The intricate interplay between glycosylation and anti-tumor immunity is a timely topic and holds great potential for future drug therapies. I do have a couple of issues that in my opinion require further evaluation by the authors.

1. In figure 1 the authors analyze the glycome of several cancer cell lines upon tumor formation in the mouse. Although this yields clear differences I am missing the $t=0$ starting point. What is the basal glycosylation of these cell lines? Does that match the tumor or are specific glycan species being induced upon tumor formation? A lectin array of the basal cell lines, grown under 3D in vitro culture conditions, is therefore needed to fully comprehend what is happening in vivo.
2. In figure 1 indeed an increase in PHA-L and PHA-E binding is observed, which the authors combinedly address as an increase in N-glycan branching. Have the authors done any MGAT3 KO studies to address the contribution of MGAT3 and does that yield similar results as for MGAT5 KO? Does 2-DG impair expression of bisecting GlcNAc and PHA-E binding?
3. I still have a problem with the use of 2-DG as a specific N-glycan modifier. Yes, the authors show that in vitro it has no effect on proliferation and glycolysis, however in vivo this might be different. Moreover, in vivo 2-DG will not only modify the cancer cells, but also affect other cell type within the tumor, such as immune cells and fibroblasts. Therefore, to pinpoint the effects truly to MGAT5 the in vivo control experiment is needed in which the MGAT5 knock down cells are used. If their hypothesis is true this should yield a similar phenotype and convey sensitivity in vivo to anti-PD-L1 treatment.

4. Have the authors ever tried anti-PD-1 blockade? This would be a very nice control experiment, as anti-PD-1 treatment should not be affected by 2-DG.

Minor points:

1. The glycan scheme included Fig. 2A contains an error that need correction: FUT7 is depicted here as generating the Lewis X structure. This is incorrect as FUT7 will only generate sialyl-Lewis X.

2. Fig. 5C is missing the PE01 (HR-deficient) control experiment.

3. In Fig. 6F a clear shift in populations/clusters is seen in the 2-DG+PD-L1 group. What are these different populations/clusters and why is there a shift upon 2-DG+PD-L1? How was this UMAP generated? This was unclear to me from the text.

Reviewer #2 (Remarks to the Author): with expertise in ovarian cancer, cancer immunology

This study by Nie, et al. investigated the difference of glycosylation between HR-proficient and HR-deficient epithelial ovarian cancer (EOC) cells using murine and human cell lines. The authors demonstrated that while fucosylation was induced by immune pressure regardless of HR-status, branched N-glycans were higher on HR-proficient cells than HR-deficient cells. Then, the authors analyzed the relationship between MGTA5 expression which is an enzyme catalyzes branched N-glycans and BRCA1/2 expression, and demonstrated BRCA1/2 regulates MGTA5 transcription. Finally, they demonstrated 2DG that is a branched N-Glycans inhibitor in combination with anti-PD-L1 enhanced CD8+ cytotoxicity and activation in vitro and in vivo. The results are interesting and worth to further investigations for EOC patients who did not have benefits from anti-PD-L1 trials. However, the manuscript has major concerns to support the conclusion and evidence.

Major comments:

1. Although the authors showed BRCA1/2 directly regulates MGAT5 expression, the correlation of MGAT5 and BRCA1/2 expression analyzed by CCLE (Fig. 2g) and TCGA (Fig. 2h)

were very weak ($\rho < 0.3$). Other mechanisms should be discussed.

2. It is not clear why 2-FF and 2-DG treatment alone did not suppress tumor growth if those compounds could inhibit PD-1/PD-L1 binding. Okada et al. (Cell Reports, 2017) reported that 2-FF treatment alone enhanced T-cell activation and anti-tumor immune responses. The treatment does not sufficient to block PD-L1 binding? In Fig. 6f, 2DG alone increased IFN-g expression compared to control, but no anti-tumor effect?

3. Some of the data are insufficient to conclude branched N-glycans regulates immune checkpoint sensitivity on HR-proficient EOC. For example, binding of PHA-L was only shown on HR-proficient cells, but not HR-deficient cells. PD-L1 expression level between HR-proficient and HR-deficient cells are not present.

Other comments:

1. Tumor volume of control and most of PD-L1 treatment group in Figure 2g-h and Figure 6b-d looks similar. The experiments were simultaneously performed for 2-DG and 2-FF treatment under the same conditions although it was not stated?

2. It is not clear why the authors focused on N-glycans not other glycans based on Fig. 1d. In Fig. 1d, PHA-E expression on HR-proficient ID8 looks similar to HR-deficient cell lines.

3. Why cytotoxicity of CAR against PEO1 was not increased by anti-PD-L1 treatment in Fig. 5g? It should behave like PEO4 with MGAT5 shRNA? The base level of cytotoxicity against different cancer cells should be included in the extended Figure.

Reviewer #3 (Remarks to the Author): with expertise in ovarian cancer, cancer immunology

Nie et al. describe a new method to treatment homologous recombination (HR)-proficient epithelial ovarian cancers (EOCs). Their main finding is that BRCA1/2 transcriptionally promotes the expression of MGAT5, the enzyme responsible for catalyzing branched N-glycans in HR-proficient tumors and augment their resistance to anti-PD-L1 by enhancing the interaction of PD-L1 and PD-1 between HR-proficient tumors and CD8+ T cells. They also show that inhibiting branched N-glycans with 2-Deoxy-D-glucose can sensitize HR-proficient, but not HR-deficient to anti-PD-L1 therapy. The findings are interesting and novel, the major conclusions are supported by their data. Overall, this study elucidates a new resistant

mechanism and may bring new insights into the treatment of HR-proficient ovarian tumors.

1. Reporter assays using the MGAT5 promoter is required to further evaluate the transcriptional regulation of BRCA1 on MGAT5 in HR-proficient tumor cells. Which domain of BRCA1 is responsible for the regulation? In Figure 4d, the level of Branched N-glycans needs to be detected in both control and BRCA1/2-knockdown cells.
2. In figure 6f and 6g, the IFN- γ expression in CD8+ T cells needs to be assessed by flow cytometric analysis. It is better to include PD-L1 and PD-1 expression analysis in the experiments.
3. BRCA1 and BRCA2 play an integral role in homologous recombination repair of double-strand DNA breaks. Is the DNA repair function of BRCA1/2 involved in regulating expression of MGAT5 in HR-proficient ovarian cancer cells?

Reviewer #4 (Remarks to the Author): with expertise in glycosylation, cancer immunology

Nie and colleagues report on the role of glycosylation in epithelial ovarian cancer cells. By using mouse models of HR-proficient and -deficient EOC in immunodeficient and -sufficient mice, the authors show increased fucosylation in immunocompetent mice and differences in branched N-glycans in HR-proficient models using a lectin array. The authors then show inhibition of fucosylation led to an improved tumor control and inhibition of N-glycan branching with 2-deoxy-d-glucose led to improved efficacy of PD-1 inhibition. This is a very interesting manuscript, although a few questions should be addressed before publication of the manuscript.

1. How is fucosylation influencing tumor growth? What are the mechanisms involved? What happens with other glycans upon 2FF treatment?
2. Do the authors observe any changes in immune cells in the tumors treated with 2FF?
3. What is effect of 2DG on other glycans? What is the effect on immune cell infiltration? Are there any changes on immune infiltrats?

4. What is the general role of N-glycans in the in vitro models? Could enzymatic treatment be used?

5. 'Targeting glycomes' is a strange title. Could the authors be more specific (fucosylation, N-glycan branching)?

We are very grateful to the reviewers for providing us with very helpful, consistent, and constructive feedback on our work, and for giving us the opportunity to revise and improve our manuscript, now entitled "*Targeting Branched N-Glycans and Fucosylation Sensitizes Ovarian Tumors to Immune Checkpoint Blockade*."

We have conducted several new *in vivo* and *in vitro* experiments, performed new analyses, and made several modifications throughout the manuscript and its figures to address all the reviewers' concerns. We have included detailed, point-by-point responses to each of these concerns, describing the corresponding changes in our manuscript. Our responses are highlighted in blue text to facilitate the review process.

Reviewer #1

Major comments:

1. *In figure 1 the authors analyze the glycome of several cancer cell lines upon tumor formation in the mouse. Although this yields clear differences I am missing the t=0 starting point. What is the basal glycosylation of these cell lines? Does that match the tumor or are specific glycan species being induced upon tumor formation? A lectin array of the basal cell lines, grown under 3D in vitro culture conditions, is therefore needed to fully comprehend what is happening in vivo.*

Authors' response: We thank the reviewer for this suggestion and we have updated **Figure 1b and 1d** to include the glycomic profiles of all cell lines *in vitro* (t=0 before injecting to the mice). Notably, the *in vitro* glycomic profiles of these cell lines differ from their *in vivo* counterparts. This supports the notion that the glycomic profiles of these cells shift upon tumor formation and under immune pressures. These findings further support our experimental approaches where we performed glycomic profiles *in vivo*.

2. *In figure 1 indeed an increase in PHA-L and PHA-E binding is observed, which the authors combinedly address as an increase in N-glycan branching. Have the authors done any MGAT3 KO studies to address the contribution of MGAT3 and does that yield similar results as for MGAT5 KO? Does 2-DG impair expression of bisecting GlcNAc and PHA-E binding?*

Authors response: Following the reviewer's suggestion, we conducted new experiments to knock down MGAT3 in HR-proficient ovarian cancer cells (OVCAR3) (**Extended Data Figure 3a**). Our new results show that MGAT3 knockdown did not decrease binding to PHA-L lectin (**Extended Data Figure 3b**) as did MGAT5 knockdown (as illustrated in Figure 3B). Additionally, we investigated the effects of 2DG on the binding of both PHA-L (**Figure 5a**) and PHA-E (**Figure 5b**), finding that 2DG treatment lessened binding of cancer cells to both lectins.

3. *I still have a problem with the use of 2-DG as a specific N-glycan modifier. Yes, the authors show that in vitro it has no effect on proliferation and glycolysis, however in vivo this might be different. Moreover, in vivo 2-DG will not only modify the cancer cells, but also affect other cell type within the tumor, such as immune cells and fibroblasts. Therefore, to pinpoint the effects truly to MGAT5 the in vivo control experiment is needed in which the MGAT5 knock down cells are used. If their hypothesis is true this should yield a similar phenotype and convey sensitivity in vivo to anti-PD-L1 treatment.*

Authors response: To isolate the effect of 2DG on other cell types, we performed new *in vivo* experiments to knock down MGAT5 in HR-proficient mouse ovarian cancer cells (KPCA). Subsequently, we injected both MGAT5 knockdown KPCA and control KPCA to induce tumor formation in immune-competent mice. As depicted in **Figure 6a**, we observed a phenotype similar to that of 2DG treatment; while MGAT5 knockdown alone did not affect tumor weight, its knockdown also sensitized KPCA tumors to anti-PD-L1 treatment, consistent with our hypothesis.

4. *Have the authors ever tried anti-PD-1 blockade? This would be a very nice control experiment, as anti-PD-1 treatment should not be affected by 2-DG.*

Authors response: Following the reviewer's suggestion, we have performed new *in vivo* experiment using mice injected with the HR-proficient mouse ovarian cancer cells (KPCA) and treated with either anti-PD-1 blockade or anti-PD-L1 blockade, in the presence or absence of 2DG. As shown in the new **Figure 7d**, while 2DG affected both anti-PD-1 blockade and anti-PD-L1 blockade, the effects were more pronounced with the

anti-PD-L1 blockade. Considering anti-PD-1 blockade is not 100% effective in blocking all PD-1/PD-L1 interactions, 2DG could enhance the effects of both anti-PD-1 and anti-PD-L1. However, as predicated by the reviewer, it should enhance anti-PD-L1 more than anti-PD-1 (which is what we observed). This is because, in the case of anti-PD-1, it only affects the residual PD-1/PD-L1 binding not blocked by anti-PD-1. Whereas with anti-PD-L1, its effects should be more generalized.

Minor comments:

1. The glycan scheme included Fig. 2A contains an error that need correction: FUT7 is depicted here as generating the Lewis X structure. This is incorrect as FUT7 will only generate sialyl-Lewis X.

Authors response: We thank the reviewer for noticing this oversight and have fixed it.

2. Fig. 5C is missing the PEO1 (HR-deficient) control experiment.

Authors response: We have now added **Extended Data Figure 7**, where we show the expression of PD-L1 and the binding to PD-1 of multiple HR-deficient and HR-proficient cells upon IFN γ stimulation. As shown, while HR-proficient cells exhibit similar levels of PD-L1 expression to HR-deficient cells (including PEO1 cells), the binding of these cells to PD-1 trends higher. This supports the notion that qualitative differences of these cells (such as glycosylation) impact the binding of PD-L1 to PD-1 and thus can affect their signaling.

3. In Fig. 6F a clear shift in populations/clusters is seen in the 2-DG+PD-L1 group. What are these different populations/clusters and why is there a shift upon 2-DG+PD-L1? How was this UMAP generated? This was unclear to me from the text.

Authors response: To address this question, we are detailing our analysis method for these data below:

“A combination of unsupervised clustering using Seurat’s K-nearest neighbour algorithm and cell-type prediction using SingleR was used to assign cell types to the immune cells. UMAPs were used for visualization of clusters (Panel A in the figure below). The T and NKT cell cluster was subset and re-clustered to further resolve T cells into CD4+ T cells (including FoxP3+ T-regs) and CD8+ T cells (memory and effector) (Panel B in the figure below). The control sample had significantly fewer T cells than others. Gene expression for IFNG was then visualized as a density plot computed using kernel densities across samples and was found to be concentrated in the CD8+ T cell region, particularly enriched in the 2DG+PDL1 sample (Panel C in the figure below).”

Reviewer #2

Major comments:

1. *Although the authors showed BRCA1/2 directly regulates MGAT5 expression, the correlation of MGAT5 and BRCA1/2 expression analyzed by CCLE (Fig. 2g) and TCGA (Fig. 2h) were very weak ($\rho < 0.3$). Other mechanisms should be discussed.*

Authors' response: To address this concern, we conducted new experiments wherein we inserted the MGAT5 promoter into the pGL4.10[luc2] vector. We then utilized the Dual-Luciferase® Reporter Assay System to determine if BRCA1 knockdown affects the expression of the Firefly Luciferase signal, which is regulated by the MGAT5 promoter in this system. We found that BRCA1 knockdown indeed significantly decreases the Firefly Luciferase signal in OVCAR3. The restoration of full-length BRCA1 can reverse this decrease, but the restoration of truncated BRCA1 without BRCT1&2 domains fails to do so. The BRCT1&2 domains are located in the C-terminus of BRCA1, functioning to regulate transcription. This suggests that MGAT5 expression is indeed regulated by BRCA1 at the transcriptional level. These data are in new **Figure 4f-g**.

Additionally, we investigated whether BRCA1's DNA repair function is involved in promoting MGAT5 expression. To activate BRCA1's DNA repair function, DNA damage was induced by cisplatin in HR-proficient OVCAR3 cells. Indeed, we observed an increase in the formation BRCA1 foci, a marker of its activation in DNA repair. Notably, MGAT5 expression is reduced in cisplatin-treated OVCAR3 cells (**Figure 4i**). Thus, BRCA1's role in promoting MGAT5 is not due to activation of its role in DNA damage repair.

2. *It is not clear why 2-FF and 2-DG treatment alone did not suppress tumor growth if those compounds could inhibit PD-1/PD-L1 binding. Okada et al. (Cell Reports, 2017) reported that 2-FF treatment alone enhanced T-cell activation and anti-tumor immune responses. The treatment does not sufficient to block PD-L1 binding? In Fig. 6f, 2DG alone increased IFN-g expression compared to control, but no anti-tumor effect?*

Authors' response: There are several key differences in our focus and approach compared to those of Okada et al., *Cell Reports*, 2017. In their study, Okada et al. focused on the fucosylation of T cells, treating them in vitro with 100 μ M of 2FF for 6 days. Subsequently, these 2FF-treated T cells were injected into tumor-bearing mice, followed by anti-PD-L1 immunotherapy, but the mice were not directly treated with 2FF. In contrast, our study focuses on the glycosylation of cancer cells themselves. We pre-treated cancer cells with 2FF for 3 days, then injected these cells into mice, and also administered 2FF directly to the mice (oral gavage at a dose of 17.5 mg/kg every 2 days). This approach, similar to that used by Yun Huang et al. in *Nature Communications*, 2021 (PMID: 33976130), where the investigators used this approach to suppress the progression of breast cancer. Our manuscript focuses on the inhibition of fucosylation of cancer cells, as opposed to the T-cell focus in Okada et al.'s study, which could account for the observed differences. Additionally, the variance in cancer models used in these studies could be another key factor, as the microenvironments and the immune inhibition caused by different tumors can significantly differ. In our study, we focused on EOCs, particularly comparing HR-proficient versus HR-deficient EOCs.

Some of the data are insufficient to conclude branched N-glycans regulates immune checkpoint sensitivity on HR-proficient EOC. For example, binding of PHA-L was only shown on HR-proficient cells, but not HR-deficient cells. PD-L1 expression level between HR-proficient and HR-deficient cells are not present.

Authors response: For the PHA-L level, Figure 1D shows that PHA-L binding is higher in two HR-proficient tumors than in two HR-deficient tumors. We have now added **Extended Data Figure 7**, where we show the expression of PD-L1 and the binding to PD-1 of multiple HR-deficient and HR-proficient cells upon IFN-gamma stimulation. As shown, while HR-proficient cells exhibit similar levels of PD-L1 expression to HR-deficient cells (including PEO1 cells), the binding of these cells to PD-1 is higher. This supports the notion that qualitative differences of these cells (such as glycosylation) impact the binding of PD-L1 to PD-1 and thus can affect their signaling.

Minor comments:

1. *Tumor volume of control and most of PD-L1 treatment group in Figure 2g-h and Figure 6b-d looks similar. The experiments were simultaneously performed for 2-DG and 2-FF treatment under the same conditions although it was not stated?*

Authors response: Yes, for each same cell lines, the experiments simultaneously performed for 2-DG and 2-FD treatment, including co-treating with anti-PD-L1. We included this information in Methods section in the initial submission and will now explicitly state this in the relevant figure legends as well.

2. *It is not clear why the authors focused on N-glycans not other glycans based on Fig. 1d. In Fig. 1d, PHA-E expression on HR-proficient ID8 looks similar to HR-deficient cell lines.*

Authors response: As the reviewer rightly pointed out we have observed several glycomic alterations caused by immune pressure on cancer cells or by differences in the phenotype of cancer cells. In this manuscript, we have focused on branched N-glycans due to their previous connection to T cell biology and their potential impact on the efficacy of anti-tumor immunotherapy. However, other differences in the glycan structures that we observed warrant further investigation in future studies, and we have made this clearer in the Discussion section.

3. *Why cytotoxicity of CAR against PEO1 was not increased by anti-PD-L1 treatment in Fig. 5g? It should behave like PEO4 with MGAT5 shRNA? The base level of cytotoxicity against different cancer cells should be included in the extended Figure.*

Authors response: The reviewer is correct. We have observed that HR-deficient PEO1 is not sensitive to T cell-mediated killing, even in the presence of 2DG and anti-PD-L1. This suggests the binding of PD-1 and PD-L1 is not the major strategy that PEO1 escape T cell-mediated killing, other receptors may be involved in this process. We have now added base level of elimination index in **Extended Data Figure 8d-f**.

Reviewer #3

Major comments:

1. *Reporter assays using the MGAT5 promoter is required to further evaluate the transcriptional regulation of BRCA1 on MGAT5 in HR-proficient tumor cells. Which domain of BRCA1 is responsible for the regulation? In Figure 4d, the level of Branched N-glycans needs to be detected in both control and BRCA1/2-knockdown cells.*

Authors response: To address this concern, we conducted new experiments wherein we inserted MGAT5 promoter into pGL4.10[luc2] Vector (Promega, E6651), then used Dual-Luciferase® Reporter Assay System (Promega, E1910) to detect if BRCA1 knockdown affect expression of Firefly Luciferase signal, which is regulated by MGAT5 promoter in this system. We found that BRCA1 knockdown indeed dramatically decrease Firefly Luciferase signal in OVCAR3. Restore full length of BRCA1 can rescue this decrease, but restore BRCT1&BRCT2 truncated BRCA1 fail to rescue this decrease. BRCT1&2 domains are in N-termination of BRCA1, which functions to regulate transcription. This indicated that MGAT5 expression is regulated by C-termination of BRCA1. These data are in new **Figure 4f-g**.

We have also added new data in **Figure 4e** showing the levels of branched N-glycans (based on binding to PHA-L) after BRCA1/2 knockdown in OVCAR3; we observed that BRCA1/2 knockdown indeed decreased branched N-glycans.

2. *In figure 6f and 6g, the IFN-γ expression in CD8+ T cells needs to be assessed by flow cytometric analysis. It is better to include PD-L1 and PD-1 expression analysis in the experiments.*

Authors response: Following the reviewer's suggestion, we have measured the IFN γ protein expression on CD8⁺ T cells by flow cytometry and these data are now in **Figure 7c**. We have also measured the expression of both PD1 and PD-L1 and added this to **Extended Data Figure 10c-d**.

3. *BRCA1 and BRCA2 play an integral role in homologous recombination repair of double-strand DNA breaks. Is the DNA repair function of BRCA1/2 involved in regulating expression of MGAT5 in HR-proficient ovarian cancer cells?*

Authors response: To address this concern, we conducted new experiments wherein we used cisplatin treatment to induce double-strand DNA breaks in OVCAR3, In immunofluorescence (IF) assay, we observed that BRCA1 form foci after cisplatin treatment, indicated BRCA1 functions to repair DNA damage at this time. Notably, MGAT5 expression is reduced in these cells (**Figure 4h**). Thus, BRCA1's role in promoting MGAT5 is not due to activation of its role in DNA damage repair.

Reviewer #4

Major comments:

1. *How is fucosylation influencing tumor growth? What are the mechanisms involved? What happens with other glycans upon 2FF treatment?*

Authors response: We thank the reviewer and have now added the following background of the potential fucosylation's role in cancer to the manuscript. "Fucosylated glycomic antigens have a significant association with cancer and immunity.¹ For instance, Lewis antigens, which contain branched fucose, on colon tumor cells, can bind to the C-type lectin DC-SIGN on macrophages and immature dendritic cells, thereby influencing the functions of these immune cells.² Another notable instance is core fucosylation, regulated by FUT8, frequently upregulated in various cancers. Disrupting this core fucosylation process can impede PD-1/PD-L1-mediated immune evasion, thereby bolstering anti-tumor immunity.³ Consequently, targeting fucosylated glycans on the glycoproteins of tumor cells has been recognized as a viable and promising strategy to amplify anti-tumor immune responses.⁴

1. Schneider, M., Al-Shareffi, E. & Haltiwanger, R. S. Biological functions of fucose in mammals. *Glycobiology* **27**, 601-618, doi:10.1093/glycob/cwx034 (2017).
2. van Gisbergen, K. P. J. M., Aarnoudse, C. A., Meijer, G. A., Geijtenbeek, T. B. H. & van Kooyk, Y. Dendritic cells recognize tumor-specific glycosylation of carcinoembryonic antigen on colorectal cancer cells through dendritic cell-specific intercellular adhesion molecule-3-grabbing nonintegrin. *Cancer Res* **65**, 5935-5944, doi:10.1158/0008-5472.Can-04-4140 (2005).
3. Liao, C. *et al.* FUT8 and Protein Core Fucosylation in Tumours: From Diagnosis to Treatment. *J Cancer* **12**, 4109-4120, doi:10.7150/jca.58268 (2021).
4. Huang, Y. *et al.* FUT8-mediated aberrant N-glycosylation of B7H3 suppresses the immune response in triple-negative breast cancer. *Nat Commun* **12**, 2672, doi:10.1038/s41467-021-22618-x (2021).

2. *Do the authors observe any changes in immune cells in the tumors treated with 2FF?*

Authors response: We have now added single-cell RNA sequencing data to **Figure 2h-i**, showing the effects of 2FF on immune cells *in vivo*.

3. *What is effect of 2DG on other glycans? What is the effect on immune cell infiltration? Are there any changes on immune infiltrats?*

Authors response: We have conducted a new experiment to examine the impact of 2DG treatment on other glycans using the lectin microarray. These data are now presented in **Extended Data Figure 6a**. In addition, we have included **Extended Data Figure 10a-b**, showing the effects of 2DG treatment *in vivo* on immune cell infiltration into tumors.

4. *What is the general role of N-glycans in the in vitro models?*

Authors response: While the role of glycans in general is emerging, with new functions in modulating various cellular processes being discovered thanks to recent technological advancements in glycobiology, a primary role of branched N-glycans has been identified. These glycans enhance the stabilization of growth factor receptors and destabilize intercellular adhesion. Consequently, *in vitro*, an increase in branched N-glycans can disrupt cell adhesion and promote metastasis. However, recent research, including ours, points to novel roles of these glycans in modulating the interactions between immune cells and cells expressing these glycans.

5. *'Targeting glycomes' is a strange title. Could the authors be more specific (fucosylation, N-glycan branching)?*

Authors response: We have now updated the title to "**Targeting Branched N-Glycans and Fucosylation Sensitizes Ovarian Tumors to Immune Checkpoint Blockade.**"

REVIEWERS' COMMENTS

Reviewer #1 (Remarks to the Author):

I thank the authors for their replies and additional experiments, which in my opinion have strengthened the manuscript.

Reviewer #2 (Remarks to the Author):

All my concerns have been appropriately addressed.

Reviewer #3 (Remarks to the Author):

All the comments have been addressed.

Reviewer #4 (Remarks to the Author):

The authors have addressed all my questions.

REVIEWERS' COMMENTS

Reviewer #1 (Remarks to the Author):

I thank the authors for their replies and additional experiments, which in my opinion have strengthened the manuscript.

Response: We thank the reviewer for the comments.

Reviewer #2 (Remarks to the Author):

All my concerns have been appropriately addressed.

Response: We thank the reviewer for the comments.

Reviewer #3 (Remarks to the Author):

All the comments have been addressed.

Response: We thank the reviewer for the comments.

Reviewer #4 (Remarks to the Author):

The authors have addressed all my questions.

Response: We thank the reviewer for the comments.